# FEDNE: Surrogate-Assisted Federated Neighbor Embedding for Dimensionality Reduction

**Ziwei Li, Xiaoqi Wang, Hong-You Chen, Han-Wei Shen, Wei-Lun Chao**
The Ohio State University
{li.5326, wang.5502, chen.9301, shen.94, chao.209}@osu.edu

## Abstract

Federated learning (FL) has rapidly evolved as a promising paradigm that enables collaborative model training across distributed participants without exchanging their local data. Despite its broad applications in fields such as computer vision, graph learning, and natural language processing, the development of a data projection model that can be effectively used to visualize data in the context of FL is crucial yet remains heavily under-explored. Neighbor embedding (NE) is an essential technique for visualizing complex high-dimensional data, but collaboratively learning a joint NE model is difficult. The key challenge lies in the objective function, as effective visualization algorithms like NE require computing loss functions among pairs of data. In this paper, we introduce FEDNE, a novel approach that integrates the FEDAVG framework with the contrastive NE technique, without any requirements of shareable data. To address the lack of inter-client repulsion which is crucial for the alignment in the global embedding space, we develop a surrogate loss function that each client learns and shares with each other. Additionally, we propose a data-mixing strategy to augment the local data, aiming to relax the problems of invisible neighbors and false neighbors constructed by the local $k$NN graphs. We conduct comprehensive experiments on both synthetic and real-world datasets. The results demonstrate that our FEDNE can effectively preserve the neighborhood data structures and enhance the alignment in the global embedding space compared to several baseline methods.

## 1 Introduction

Federated Learning (FL) has emerged as a highly effective decentralized learning framework in which multiple participants collaborate to learn a shared model without sharing the data. In recent years, FL has been extensively studied and applied across various domains, including image and text classifications [10, 31, 23, 9], computer vision tasks [3, 41, 50], and graph learning problems [42, 5, 16]. Despite the growing interest in FL, the area of dimensionality reduction within this framework has received limited investigation. However, visualizing and interpreting data from distributed sources is important, as real-world applications often generate large volumes of complex datasets that are stored locally by each participant. For example, different hospitals collect high-dimensional electronic health records (EHRs) [1] and securely store this patient data within their local systems. As each hospital might only collect limited data or focus on particular diseases, conducting data visualization on a combined dataset can substantially improve disease diagnosis and provide deeper insights. However, sharing sensitive patient information is restricted due to privacy protection. Thus, developing a shared dimensionality reduction model in the FL setting is crucial for facilitating collaborative analysis while maintaining data on local sites.

Dimensionality reduction (DR) refers to constructing a low-dimensional representation from the input data while preserving the essential data structures and patterns. Neighbor embedding (NE) [6, 49], a family of DR techniques, is widely utilized to visualize complex high-dimensional data due to its ability to preserve neighborhood structures and handle non-linear relationships effectively. Essentially,

38th Conference on Neural Information Processing Systems (NeurIPS 2024).

NE methods (e.g., t-SNE [44, 43] and UMAP [33]) operate on an attraction-repulsion spectrum [6], balancing attractive forces that pull similar data points together and repulsive forces that push dissimilar points apart. Defining the objective function requires access to any pairs of data points.

Such a need to access pairwise data for determining the attraction and repulsion terms, however, poses critical challenges to the FL framework. As data are distributed across different clients, computing their pairwise distances becomes non-trivial, making it difficult to recover the centralized objective in an FL setting. Specifically, the *absence of inter-client repulsive forces* complicates the separation of dissimilar data points. Moreover, within a client, due to the unavailability of others' data, *defining a faithful attraction term based on the top neighbors is challenging*, often resulting in the inaccurate grouping of distant data points. This contrasts with traditional FL tasks, such as image classification, where learning objectives can be decomposed over individual training instances, allowing each client to calculate the loss independently and optimize the model based solely on its local dataset.

To the best of our knowledge, only a few existing works address the problem of decentralized data visualization. Both dSNE [37] and F-dSNE [38] methods require publicly shared data that serves as a reference point for aligning the embeddings from different clients. This setup introduces additional assumptions that may not be feasible in real-world applications, and the quantity and representativeness of the reference data can significantly impact the resulting embeddings.

To this end, we proposed a novel Federated neighbor embedding technique, called FEDNE, which follows the widely used FEDAVG pipeline. It trains a shared NE model that is aware of the global data distribution, without requiring any shareable data across participants. To compensate for the lack of much-needed inter-client repulsive force, besides training a local copy of the NE model, each client additionally learns a surrogate model designed to summarize its local repulsive loss function. During global aggregation, this surrogate model will be sent back to the server along with the local NE model, which other clients can use in the next round of local training. In detail, for a client $m$, its local surrogate model is designed to approximate the repulsion loss from an arbitrary point to its local data points. By sending the surrogate model to other clients, another client $m'$ can incorporate it into its local loss function for training the NE model. Additionally, to handle the difficulty of estimating the neighborhood, we introduce an intra-client data mixing strategy to simulate the presence of potential neighbors residing on the other clients. This approach augments the local data to enhance the training of the NE model.

To showcase the effectiveness of FEDNE, we conduct comprehensive experiments using both synthetic and real-world benchmark datasets used in the field of dimensionality reduction under various FL settings. Both qualitative and quantitative evaluation results have demonstrated that our method outperforms the baseline approaches in preserving the neighborhood data structures and facilitating the embedding alignment in the global space.

**Remark**. It is worth discussing that we understand privacy-preserving is an important aspect to address in the FL framework. However, we want to reiterate the main focus of this paper is identifying the unique challenges in the federated neighbor embedding problem and proposing effective solutions rather than resolving all the FL challenges at once. We discuss the privacy considerations and our further work in section 6.

## 2 Related Work

**Federated learning.** FL aims to train a shared model among multiple participants while ensuring the privacy of each local dataset. FEDAVG [34] is the foundational algorithm that established the general framework for FL. Subsequent algorithms have been proposed to further improve FEDAVG in terms of efficiency and accuracy. Some of the work focuses on developing advanced aggregation techniques from various perspectives such as distillation [46, 39], model ensemble [30, 40], and weight matching [45, 52] to better incorporate knowledge learned by local models. Moreover, to minimize the deviation of local models from the global model, many works focus on enhancing the local training procedures [21, 2, 51, 29]. FEDXL [15] was proposed as a novel FL problem framework for optimizing a family of risk optimization problems via an active-passive decomposition strategy. Even though FEDXL deals with the loss decomposition for pairwise relations, our main focus and application are very different.

**Neighbor embedding.** Neighbor embedding (NE) is a family of non-linear dimensionality reduction techniques that rely on $k$-nearest neighbor ($k$NN) graphs to construct the neighboring relationships

within the dataset [6]. The key of NE methods lies in leveraging the interplay between attractive forces which bring neighboring data points closer and repulsive forces which push uniformly sampled non-neighboring data pairs further apart. t-SNE [44] is a well-known NE algorithm. It first converts the data similarities to joint probabilities and then minimizes the Kullback–Leibler divergence between the joint probabilities of data pairs in the high-dimensional space and low-dimensional embedding space. Compared to t-SNE, UMAP [33] is better at preserving global data structure and more efficient in handling large datasets. A later study has analyzed the effective loss of UMAP[13] and demonstrated that the negative sampling strategy indeed largely reduces the repulsion shown in the original UMAP paper, which explains the reasons for the success of UMAP. Our federated NE work is built upon a recent work that has theoretically connected NE methods with contrastive loss [12, 19]. Their proposed framework unifies t-SNE and UMAP as a spectrum of contrastive neighbor embedding methods.

**Decentralized dimensionality reduction.** As nowadays datasets are often distributively stored, jointly analyzing the data from multiple sources has become increasingly important especially when the data contains sensitive information. SMAP [47] is a secure multi-party t-SNE. However, as this framework requires data encryption, decryption, and calculations on the encrypted data, SMAP is very time-consuming and thereby it can be impractical to run in real-world applications. dSNE was proposed for visualizing the distributed neuroimaging data [37]. It assumes that a public neuroimaging dataset is available to share with all participants. The shareable data points act as anchors for aligning the local embeddings. To improve the privacy and efficiency of dSNE, Faster AdaCliP dSNE (F-dSNE) [38] was proposed with differential privacy to provide formal guarantees. While their goal is not to collaboratively learn a global predictive DR model and thus does not follow the formal FL protocols [22, 8, 34] defined in the literature. Both methods require a publicly available dataset to serve as reference gradients communicating across central and local sites. However, since a public dataset may not be available in most real-world scenarios, our FEDNE is designed without any requirements for the shareable data.

## 3 FL Framework for Neighbor Embedding

In this section, we first provide background information on neighbor embedding techniques. We then formulate the problem within the context of FL and outline the unique challenges.

### 3.1 Contrastive Neighbor Embedding

The goal of general NE techniques is to construct the low-dimensional embedding vectors $z_1, ..., z_N \in \mathbb{R}^d$ for input data points $x_1, ..., x_N \in \mathbb{R}^D$ that preserve pairwise affinities of data points in the high-dimensional space. The neighborhood relationships are defined via building sparse k-nearest-neighbor ($k$NN) graphs over the entire dataset with a fixed $k$ value. Contrastive NE [11] is a unified framework that establishes a clear mathematical relationship among a range of NE techniques including t-SNE [44], UMAP [13], and NCVis [4], via contrastive loss. For parametric NE, an encoder network $f_\theta$ is trained to map an input data point $x$ to a low-dimensional representation $z$, i.e., $z = f_\theta(x)$.

In general, the contrastive NE algorithms first build $k$NN graphs to determine the set of neighbors $p_i$ for each data point $x_i$ in the high-dimensional space. A numerically stable Cauchy kernel is used for converting a pairwise low-dimensional Euclidean distance to a similarity measure: $\phi(z_i, z_j) = \frac{1}{1+\|z_i - z_j\|_2^2}$. Then, the contrastive NE [11] loss is optimized via the negative sampling strategy:

$$\mathcal{L}(\theta) = - \underbrace{\mathbb{E}_{ij \sim p_i} \log(\phi(f_\theta(x_i), f_\theta(x_j)))}_{\text{Attractive force}} - \underbrace{b \mathbb{E}_{ij} \log(1 - \phi(f_\theta(x_i), f_\theta(x_j)))}_{\text{Repulsive force}}, \quad (1)$$

where $p_i$ denotes the set of neighboring data points of $x_i$.

### 3.2 Problem Formulation

In general federated learning with one central server and $M$ clients, each client holds its own training data $\mathcal{D}_m = \{x_i\}_{i=1}^{|\mathcal{D}_m|}$ and we denote the collective global data as $\mathcal{D}_{\text{glob}}$. The clients' datasets are disjoint which cannot be shared across different local sites, i.e., $D_m \cap D_{m'} = \emptyset$ for $\forall\, m, m' \in [M]$, and $m \neq m'$. Our goal is to learn a single neighbor embedding model such that the high-dimensional affinities of the global data can be retained in a global low-dimensional embedding (2D) space.

It is natural to consider employing the FEDAVG [34] framework since the clients can collaborate by communicating their parametric NE models. Then, the learning objective can be formulated with $\boldsymbol{x}_i, \boldsymbol{x}_j \in \mathcal{D}_m$ as following:

$$\boldsymbol{\theta}^* = \arg\min_{\boldsymbol{\theta}} \ \bar{\mathcal{L}}(\boldsymbol{\theta}) = \sum_{m=1}^{M} \frac{|\mathcal{D}_m|}{|\mathcal{D}|} \mathcal{L}_m(\boldsymbol{\theta}), \tag{2}$$

$$\text{where } \mathcal{L}_m(\boldsymbol{\theta}) = -\underbrace{\mathbb{E}_{ij \sim p_i} \log(\phi(f_\theta(\boldsymbol{x}_i), f_\theta(\boldsymbol{x}_j)))}_{\text{Attractive force}} - \underbrace{b\mathbb{E}_{ij} \log(1 - \phi(f_\theta(\boldsymbol{x}_i), f_\theta(\boldsymbol{x}_j)))}_{\text{Repulsive force}}. \tag{3}$$

As the client data cannot leave its own device, Equation 2 cannot be solved directly. The vanilla FEDAVG relaxes Equation 2 through $T$ iterations of local training and global model aggregations. The fundamental procedures are defined below,

$$\textbf{Local: } \boldsymbol{\theta}_m^{(t)} = \arg\min_{\boldsymbol{\theta}} \mathcal{L}_m(\boldsymbol{\theta}), \text{ initialized with } \bar{\boldsymbol{\theta}}^{(t-1)};$$

$$\textbf{Global: } \bar{\boldsymbol{\theta}}^{(t)} \leftarrow \sum_{m=1}^{M} \frac{|\mathcal{D}_m|}{|\mathcal{D}|} \boldsymbol{\theta}_m^{(t)}. \tag{4}$$

During local training, each participating client $m$ updates its model parameter $\boldsymbol{\theta}_m$ for only a few epochs based on the aggregated model $\bar{\boldsymbol{\theta}}^{(t-1)}$ received from the server.

### 3.3 Challenges of Federated Neighbor Embedding

However, besides the challenges posed by the non-IID data, simply decomposing the problem into Equation 3 indeed *overlooks the pairwise data relationships existing across different clients*. The major difference between the existing FL studies, e.g., image classification, and Federated neighbor embedding is that the objective function of the former problems is instance-based, where their empirical risk is the sum of the risk from each data point:

$$\mathcal{L}_m(\boldsymbol{\theta}) = \frac{1}{|\mathcal{D}_m|} \sum_{i}^{|\mathcal{D}_m|} \ell(\boldsymbol{x}_i, \boldsymbol{y}_i; \boldsymbol{\theta}). \tag{5}$$

As a result, the FL objective in Equation 2, i.e., $\sum_{m=1}^{M} \frac{|\mathcal{D}_m|}{|\mathcal{D}|} \mathcal{L}_m(\boldsymbol{\theta})$, is exactly the one as if all the clients' data were gathered at the server.

In the context of Federated neighbor embedding, Equation 3 only considers $\boldsymbol{x}_j$ to come from the same client as $\boldsymbol{x}_i$. Thus, simply adopting the vanilla FEDAVG framework can result in losing all the attractive and repulsive terms that should be computed between different clients.

Therefore, we redefine the FL objective of the contrastive neighbor embedding problem to be

$$\mathcal{L}(\boldsymbol{\theta}) = \underbrace{\sum_{m=1}^{M} \mathbb{E}_{(i,j) \sim D_m} [\ell(\boldsymbol{x}_i, \boldsymbol{x}_j; \boldsymbol{\theta})]}_{\text{Intra-client terms}} + \underbrace{\sum_{m=1}^{M} \sum_{\substack{m'=1 \\ m' \neq m}}^{M} \mathbb{E}_{(i,j) \sim (D_m, D_{m'})} [\ell(\boldsymbol{x}_i, \boldsymbol{x}_j; \boldsymbol{\theta})]}_{\text{Inter-client terms}}, \tag{6}$$

where the *pairwise* empirical risk $\ell(\boldsymbol{x}_i, \boldsymbol{x}_j; \theta)$ can be further defined as

$$\ell(\boldsymbol{x}_i, \boldsymbol{x}_j; \theta) = -\underbrace{\mathbb{1}[x_j \in p_i] \log(\phi(f_\theta(\boldsymbol{x}_i), f_\theta(\boldsymbol{x}_j)))}_{\text{Attractive force}} - \underbrace{b \log(1 - \phi(f_\theta(\boldsymbol{x}_i), f_\theta(\boldsymbol{x}_j)))}_{\text{Repulsive force}}. \tag{7}$$

Nonetheless, since the inter-client pairwise distances are unknown, Equation 7 cannot be solved directly when $\boldsymbol{x}_i$ and $\boldsymbol{x}_j$ come from different clients. Specifically, this decentralized setting brings two technical challenges: (1) *Biased repulsion loss*: Negative sampling requires selecting non-neighbor pairs uniformly across the entire data space. Under the FL setting, it is difficult for a client to sample from outside of its local dataset. (2) *Incorrect attraction loss*: Each client only has access to its local data points. This partitioning can result in an incomplete $k$NN graph, leading to incorrect $p_i$, as some true neighbors of a data point might reside on other clients.

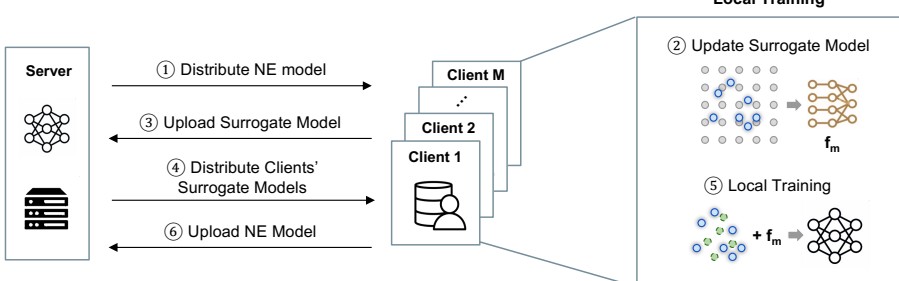

Figure 1: An illustration of one round of FEDNE. Besides the general steps in FEDAVG: ① → ⑤ → ⑥, our local surrogate model training (②) can be smoothly integrated into the whole pipeline. Then, each client conducts its local training (⑤) using the augmented data and the surrogate models received from all the other clients (③ → ④).

## 4 Federated Neighbor Embedding: FEDNE

To address the aforementioned challenges in applying the FL framework to the neighbor embedding problem, we develop a learnable surrogate loss function[1] trained by each client and an intra-client data augmentation technique to tackle the problems in repulsion and attraction terms separately. The two components can be smoothly integrated into the traditional FEDAVG pipeline shown in Figure 1.

### 4.1 Surrogate Loss Function

The repulsive force plays an important role in ensuring separation among dissimilar data points, contributing to the global data arrangement in the embedding space. In the centralized scenario, each data point can uniformly sample its non-neighbors across the entire dataset. In contrast, in the federated setting, as each client can only access its local data, the dissimilar points residing in other clients are invisible, and all the repulsion will be estimated within its own data space. In particular, under severe non-IID situations, where the data distributions across different clients vary significantly [54], the non-neighboring samples selected to repel are very likely to come from the same clusters in high-dimensional space. As showcased in Figure 2 (a), without explicitly taking care of the repulsion between dissimilar data pairs across clients, those points may still overlap with each other in the embedding space.

At a high level, we seek to learn a function $f_{m;w}^{\text{rep}} : d \rightarrow R$ for each client $m$ such that $f_{m;w}^{\text{rep}} \approx b \log(1 - \phi(f_\theta(\boldsymbol{x}_i), f_\theta(\boldsymbol{x}_j)))$ to estimate the repulsion, where $x_i$ and $x_j$ come from different clients. This surrogate model, once shared, enables other clients to input their local data points and obtain a pre-estimated repulsive loss to data points from the originating client.

**Surrogate model training.** We learn $f_{m;w}^{\text{rep}}$ via supervised learning at each round of FEDAVG. To do so, we generate a set of low-dimensional query points as inputs and pre-compute their corresponding repulsive loss to client $m$'s data points based on the current projection model. We choose to sample a set of points $Z_q$ within 2D space for the following two reasons. Firstly, as non-neighboring points are uniformly selected across the data space, query points are not required to maintain any specific affinity with $\mathcal{D}_m$. Second, because the high-dimensional space is often much sparser than 2D space, generating sufficient high-dimensional samples to comprehensively represent the data distributions of all other clients is impractical. Therefore, each client employs a grid sampling strategy at every round, using a predefined step size and extensive ranges along the two dimensions. This procedure is informed by client $m$'s current embedding positions, ensuring a more manageable and representative sampling process within the embedding space.

In sum, given the sampled inputs $Z_q = \{z_{q_1}, z_{q_2}, \ldots, z_{q_{N_q}}\}$, we prepare the training targets by computing the repulsive loss between each $z_{q_i}$ and $b$ random data points in $\mathcal{D}_m$, i.e., $l_{q_i}^{\text{rep}} = -\sum_j^b \log(1 - \phi(z_{q_i}, z_m^{(j)}))$. Then, the dataset for supervised training the surrogate repulsion function $f_{m;w}^{\text{rep}}$ is constructed as $\mathcal{D}_q = \{(z_{q_i}, l_{q_i}^{\text{rep}})\}_{i=1}^{|\mathcal{D}_q|}$.

---

[1]We use surrogate loss function and surrogate model interchangeably.

**Implementation details.** After building the training dataset, each client trains its surrogate model $f_{m;w}^{\text{rep}}$ using an MLP with one hidden layer to learn the mapping between the input embedding vectors and their corresponding repulsive loss measured within the client data by minimizing the mean squared error (MSE). The training objective is formulated as follows:

$$w^* = \arg\min_w \frac{1}{|D_q|} \sum_{i=1}^{|D_q|} \left( l_{q_i}^{\text{rep}} - f_{m;w}^{\text{rep}}(z_{q_i}) \right)^2 \tag{8}$$

## 4.2 Neighboring Data Augmentation

To mitigate the limitations of biased local $k$NN graphs and ensure better neighborhood representation, we propose an intra-client data mixing strategy. This approach generates additional neighboring data points within each client, thereby enhancing the local data diversity. To be specific, locally constructed $k$NN graphs can be biased by the client's local data distribution. As the associated data pairs for computing the attractive loss are distributed across multiple clients, the local neighborhood within each client can be very sparse. Consequently, data points within a client may miss some of their true neighbors (i.e., *invisible neighbors*) considered in the global space. Moreover, when the local data is extremely imbalanced compared to the global view, constructing the $k$NN graph with a fixed $k$ value may result in incorrect neighbor connections between very distant data points (i.e., *false neighbors*). As demonstrated in Figure 2 (b), since data is partitioned across different clients, with a fixed $k$ value, each local $k$NN graph can be even more sparse and erroneously connect very distinct data points.

**Intra-client data mixing.** To address these problems, we employ a straightforward yet effective strategy, intra-client data mixing, to locally generate some data within a client by interpolating between data points and their neighbors. Our method shares a similar spirit to the mixup augmentation [53, 36]. In detail, given a data point $\boldsymbol{x}_i$ and the set of its $k$ nearest neighbor $\text{NN}_k(\boldsymbol{x}_i)$, a new data point is generated by linearly interpolating $\boldsymbol{x}_i$ and a random sample in $\text{NN}_k(\boldsymbol{x}_i)$ denoted as $\boldsymbol{x}_j$:

$$\hat{\boldsymbol{x}} = \lambda \boldsymbol{x}_i + (1 - \lambda) \boldsymbol{x}_j, \tag{9}$$

where $\lambda$ is the weight sampled from the Beta distribution i.e., $\lambda \sim \text{Beta}(\alpha)$.

## 4.3 Overall Framework

Once each client has received the surrogate loss functions of all the other participants (i.e., step 4 in Figure 1), it proceeds to its local training. By combining the original NE loss with the surrogate loss function on the augmented local training data, the new objective for client $m$ can be formulated as:

$$\mathcal{L}_m(\hat{\mathcal{D}_m}; \theta) = \underbrace{-\sum_{ij \sim p} \log(\phi(f_\theta(\boldsymbol{x}_i^m), f_\theta(\boldsymbol{x}_j^m)))}_{\text{Original attractive loss}} \underbrace{- \frac{|\mathcal{D}_m|}{|\mathcal{D}|} \sum_{ij} \log(1 - \phi(f_\theta(\boldsymbol{x}_i^m), f_\theta(\boldsymbol{x}_j^m)))}_{\text{Original repulsive loss}}$$

$$+ \sum_{m' \neq m} \frac{|\mathcal{D}_{m'}|}{|\mathcal{D}|} \underbrace{\sum_i f_{m';w}^{rep}(f_\theta(\boldsymbol{x}_i^m))}_{\text{Surrogate repulsion loss from client } m'}, \tag{10}$$

where $\boldsymbol{x}_i^m, \boldsymbol{x}_j^m \in \hat{\mathcal{D}_m}$ i.e., the augmented training set. For simplicity, we use $p$ to represent the neighbor set constructed within $\hat{\mathcal{D}_m}$. $f_{m';w}^{rep}$ is the surrogate model received from another client $m'$.

**Computation.** At first glance, FEDNE may seem to introduce heavy computational overhead compared to the original FEDAVG framework, as it requires additional surrogate model training at every round. Moreover, a client needs to use multiple received surrogate models to do inference using its own local data. Nevertheless, we want to emphasize that the surrogate model contains only one hidden layer and takes *2D data points* as inputs. Therefore, training and using them is manageable. We conducted experiments using MNIST with 20 clients on a server with 4 NVIDIA GeForce RTX 2080 Ti GPUs. Compared to FEDAVG, our FEDNE takes $35\%$ more GPU time to complete one round of training. For future speed-up, we may consider applying strategies such as clustered FL and we leave this for future work.

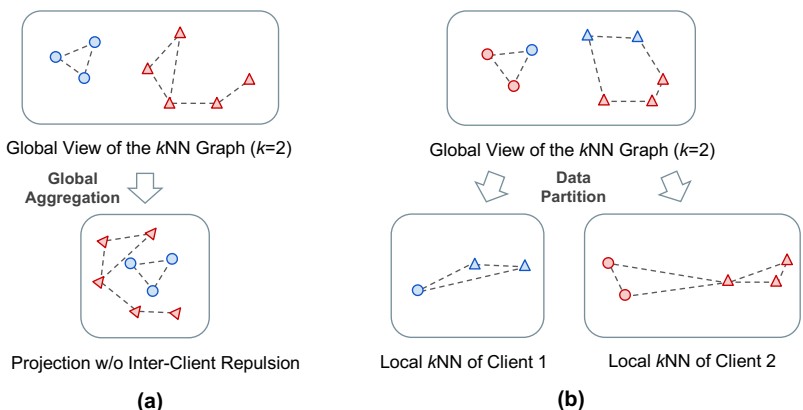

Figure 2: Toy examples for illustrating the major challenges in solving Federated NE. Color denotes the client identity, and different shapes represent the true categories of the data points (just for demonstration purposes). (a) Without repelling the dissimilar data from other clients, the projected data points from different clients may overlap with each other in the global embedding space. (b) Biased local $k$NN graphs may incorrectly connect distant data pairs as neighbors.

## 5 Experiments

### 5.1 Experimental Settings

**Datasets.** We conduct experimental studies on four benchmark datasets that have been widely used in the field of dimensionality reduction [35, 55]: MNIST [26], Fashion-MNIST [48], mouse retina single-cell transcriptomes [32], and CIFAR-10 [25]. Their statistical information and general settings are summarized in Table 1. For CIFAR-10, since the Euclidean distances in the pixel space of a natural image dataset are not meaningful to preserve [7], we adopted ImageNet-pretrained ResNet-34 [17] to extract a set of 512D feature vectors as input data. The resulting vectors still retain category-relevant structures that can be suitable for the Euclidean metric.

**Non-IID data partition.** We consider two partitioning strategies to simulate the heterogeneous client distributions: (1) *Dirichlet*: For a class $c$, we sample $q_c$ from $Dir(\alpha)$ and assign data samples of that class $c$ to a client $m$ proportionally to $q_c[m]$. The hyperparameter $\alpha$ controls the imbalance level of the data partition where a smaller $\alpha$ indicates a more severe non-IID condition [27, 18]. (2) *Shards*: each client holds data from $C$ classes, and all samples from the same class are randomly and equally divided among all clients [27].

**Evaluation metrics.** We assess the quality of data embeddings by analyzing the input high-dimensional data points and their corresponding 2D positions [14]. First, to evaluate the preservation of neighborhood structures, we compute trustworthiness and continuity scores. Trustworthiness quantifies the quality of a low-dimensional embedding by checking whether neighbors in the high-dimensional space remain the same as the ones in the embedded low-dimensional space. Conversely, continuity verifies whether the neighbors in the embedded space correspond to neighbors in the original input space. We use $k$NN classification accuracy to measure the effectiveness in preserving the nearest neighbors in the embedded space, where higher scores indicate better class discrimination. We fix $k = 7$ for all the neighborhood metrics. Additionally, we employ steadiness and cohesiveness metrics to evaluate the reliability of the global inter-cluster structures [20]. Steadiness assesses the presence of false groups, while cohesiveness checks for the existence of any missing groups.

**Implementation and training details.** We employ a fully connected neural network with three hidden layers for MNIST, Fashion-MNIST, and CIFAR-10 datasets and a network with two hidden layers for the RNA-Seq dataset. In all experiments, we use Adam optimizer [24] with learning rate annealing and a batch size of 512 where the batch size refers to the number of edges in the constructed $k$NN graphs. Furthermore, we assume full participation during the federated learning and each client performs one epoch of local training ($E = 1$). In addition, we set $\alpha = 0.2$ to perform the intra-client data augmentation in our study.

**Baselines.** We consider four approaches to compare with our FEDNE. (1) LocalNE: each client trains the NE model using only its local data without any communication. Two baseline methods:

(2) FedAvg+NE and (3) FedProx+NE are implemented by applying the widely used FL frameworks [34, 28] to NE model training. (4) GlobalNE: the NE model trained using aggregated global data, serving as the upper bound for performance comparison. Moreover, we want to emphasize that we do not include dSNE and F-dSNE for comparison. Although, at first glance, their titles might imply that they tackled a similar problem, their method is built upon the *non-parametric* t-SNE and *heavily* relies on the shareable reference dataset. Thus, they are not comparable with our framework.

Table 1: Dataset statistics and learning setups.

| Dataset | #Class | #Training | #Test | #Clients($M$) | #Dimension |
|---|---|---|---|---|---|
| MNIST | 10 | 60K | 10K | 20/100 | 784 |
| Fashion-MNIST | 10 | 60K | 10K | 20/100 | 784 |
| scRNA-Seq | 12 | 30K | 4.4K | 20/50 | 50 |
| CIFAR-10 | 10 | 50K | 10K | 20/100 | 512 |

## 5.2 Results

We conduct comprehensive experiments under various non-IID conditions and then evaluate on the global test data of each four datasets. For the highly imbalanced scRNA-Seq dataset, we only consider the Dirichlet partitions. The results of partitions under Dirichlet distributions are summarized in Table 2. Overall, our FEDNE outperforms the LocalNE by 2.62%, 6.12%, 14.31% 12.69%, and 7.31% on average under the five metrics (i.e., conti., trust., $k$NN acc., stead., and cohes.) respectively. In addition, the results of the Shards setting can be found in the appendix, i.e., Table 9.

**Improved preservation of true neighbors.** Both FEDNE and the baseline approaches achieved relatively high continuity scores, indicating that the models can easily learn how to pull the data points that are similar in the high-dimensional space closer in the 2D space. However, the lower trustworthiness scores observed with the two baselines, FedAvg+NE and FedProx+NE, imply that without properly addressing incorrect neighborhood connections and separation of data points across different clients, the resulting embeddings may contain false neighbors. Consequently, points that are positioned closely in the 2D space might not be neighbors in the original high-dimensional space.

**Enhanced class discrimination in the embedding space.** Our method has significantly improved the $k$NN classification accuracy compared to the baseline results. This improvement highlights the limitations of locally constructed $k$-NN graphs, which may incorrectly pull distant data pairs closer in the embedding space. In particular, if two data points from different classes are mistakenly treated as neighbors, class separation will be largely reduced even when inter-client repulsion is considered. Our intra-client data mixing method is specifically designed to relax this problem, and when combined with our surrogate loss function, it ensures an enhanced class separation. For instance, the embedding visualization of FedAvg+NE in Figure 3 under the $Dir(0.1)$ setup shows a significant overlap between points from different labels. In contrast, FEDNE effectively separates the top groups of features in the visualization.

**Better preservation of global inter-cluster structures.** Furthermore, we observe large improvements in preserving the clustering structures according to measures of steadiness and cohesiveness. Specifically, higher steadiness achieved by FEDNE indicates that the clusters identified in the projected space better align with the true clusters in the original high-dimensional space. The higher cohesiveness scores imply that the clusters in the high-dimensional space in general can be retained in the projected space, i.e., not splitting into multiple parts. Overall, even though FEDNE is not explicitly designed to improve feature clustering, it can produce relatively reliable inter-cluster structures.

## 5.3 Ablation Study

To verify the effect of our design choices, we conduct ablation studies on removing one of the proposed technical components from the FEDNE pipeline in Figure 1. First, we remove the data augmentation by intra-client data mixing technique and only keep the surrogate repulsion model. We then remove the surrogate model and only augment the local data using the intra-client data mixing approach. The comparison results under the setup of $Dir(0.1)$ with 20 clients are shown in Table 3. With any of the components removed, our FEDNE can still outperform the baseline FedAvg+NE. However, we cannot simply conclude which component impacts the most on the final embedding results since the data characteristics and client partitions are very different across different setups. Further studies on our design choices are included in the appendix.

Table 2: Quality of the global test 2D embedding under the non-IID Dirichlet distribution ($Dir(0.1)$ and $Dir(0.5)$) on four datasets. FEDNE achieves top performance on preserving both neighborhood structures (i.e., continuity, trustworthiness, and $k$NN classification accuracy) and global inter-cluster structures (i.e., steadiness and cohesiveness).

| Metric | Method | MNIST $M=20$ 0.1 | 0.5 | $M=100$ 0.1 | 0.5 | Fashion-MNIST $M=20$ 0.1 | 0.5 | $M=100$ 0.1 | 0.5 | RNA-Seq $M=20$ 0.1 | 0.5 | $M=50$ 0.1 | 0.5 | CIFAR-10 $M=20$ 0.1 | 0.5 | $M=100$ 0.1 | 0.5 |
|---|---|---|---|---|---|---|---|---|---|---|---|---|---|---|---|---|---|
| Conti. | LocalNE | 0.91 | 0.95 | 0.93 | 0.95 | 0.96 | 0.98 | 0.97 | 0.98 | 0.95 | 0.97 | 0.96 | 0.97 | 0.87 | 0.92 | 0.87 | 0.91 |
| | FedAvg+NE | **0.97** | **0.98** | **0.96** | **0.97** | 0.98 | **0.99** | **0.99** | **0.99** | **0.97** | **0.98** | **0.97** | **0.98** | **0.93** | **0.94** | 0.93 | 0.94 |
| | FedProx+NE | **0.97** | **0.98** | **0.96** | **0.97** | **0.99** | **0.99** | 0.98 | **0.99** | **0.97** | **0.98** | **0.97** | **0.98** | **0.93** | **0.94** | **0.94** | **0.95** |
| | FEDNE | **0.97** | 0.97 | **0.96** | **0.97** | **0.99** | **0.99** | **0.99** | **0.99** | **0.97** | **0.98** | **0.97** | **0.98** | **0.93** | **0.94** | 0.93 | 0.94 |
| | GlobalNE | 0.97 | | | | 0.99 | | | | 0.98 | | | | 0.95 | | | |
| Trust. | LocalNE | 0.75 | 0.84 | 0.74 | 0.81 | 0.89 | 0.94 | 0.89 | 0.93 | 0.80 | 0.86 | 0.79 | 0.86 | 0.74 | 0.81 | 0.73 | 0.79 |
| | FedAvg+NE | 0.78 | 0.91 | 0.74 | 0.88 | **0.95** | **0.97** | 0.89 | **0.96** | 0.85 | 0.90 | 0.84 | 0.89 | 0.82 | **0.86** | 0.78 | 0.84 |
| | FedProx+NE | 0.78 | 0.91 | 0.75 | 0.88 | **0.95** | **0.97** | 0.89 | **0.96** | 0.84 | 0.90 | 0.83 | 0.89 | 0.81 | **0.86** | **0.80** | **0.85** |
| | FEDNE | **0.85** | **0.93** | **0.82** | **0.90** | **0.95** | **0.97** | **0.95** | **0.96** | **0.87** | **0.91** | **0.85** | **0.91** | **0.83** | **0.86** | **0.80** | **0.85** |
| | GlobalNE | 0.94 | | | | 0.97 | | | | 0.93 | | | | 0.87 | | | |
| $k$NN | LocalNE | 0.44 | 0.66 | 0.43 | 0.58 | 0.53 | 0.64 | 0.53 | 0.60 | 0.81 | 0.89 | 0.80 | 0.89 | 0.44 | 0.58 | 0.43 | 0.55 |
| | FedAvg+NE | 0.48 | 0.76 | 0.43 | 0.67 | 0.60 | **0.70** | 0.55 | 0.66 | 0.85 | 0.94 | 0.84 | 0.93 | 0.55 | 0.72 | 0.48 | 0.68 |
| | FedProx+NE | 0.49 | 0.75 | 0.44 | 0.68 | 0.60 | 0.69 | 0.54 | 0.66 | 0.83 | 0.94 | 0.83 | 0.93 | 0.55 | 0.71 | 0.50 | 0.69 |
| | FEDNE | **0.72** | **0.89** | **0.65** | **0.78** | **0.66** | **0.70** | **0.66** | **0.67** | **0.90** | **0.96** | **0.88** | **0.95** | **0.63** | **0.77** | **0.54** | **0.73** |
| | GlobalNE | 0.93 | | | | 0.73 | | | | 0.97 | | | | 0.78 | | | |
| Stead. | LocalNE | 0.45 | 0.60 | 0.46 | 0.56 | 0.64 | 0.76 | 0.63 | 0.72 | 0.55 | 0.68 | 0.55 | 0.71 | 0.57 | 0.65 | 0.56 | 0.64 |
| | FedAvg+NE | 0.54 | 0.73 | 0.43 | 0.69 | 0.79 | **0.84** | 0.62 | **0.81** | 0.64 | 0.79 | 0.68 | **0.78** | **0.67** | 0.71 | 0.63 | 0.69 |
| | FedProx+NE | 0.51 | 0.72 | 0.43 | 0.68 | 0.79 | **0.84** | 0.62 | **0.81** | 0.59 | 0.79 | 0.66 | **0.78** | 0.66 | 0.71 | 0.63 | **0.71** |
| | FEDNE | **0.63** | **0.74** | **0.57** | **0.73** | **0.81** | 0.83 | **0.81** | **0.81** | **0.73** | **0.81** | **0.70** | **0.78** | **0.67** | **0.72** | **0.64** | **0.71** |
| | GlobalNE | 0.76 | | | | 0.83 | | | | 0.81 | | | | 0.73 | | | |
| Cohes. | LocalNE | 0.70 | 0.81 | 0.70 | 0.77 | 0.62 | 0.69 | 0.62 | 0.68 | 0.61 | 0.66 | 0.60 | 0.66 | 0.65 | 0.72 | 0.64 | 0.70 |
| | FedAvg+NE | 0.77 | 0.89 | 0.74 | 0.84 | 0.71 | **0.74** | 0.68 | 0.72 | 0.68 | 0.67 | **0.68** | 0.70 | 0.72 | **0.77** | **0.70** | 0.75 |
| | FedProx+NE | 0.77 | 0.89 | 0.76 | 0.86 | 0.71 | 0.73 | 0.69 | 0.73 | 0.68 | **0.68** | 0.65 | 0.69 | 0.71 | 0.76 | **0.70** | **0.76** |
| | FEDNE | **0.82** | **0.90** | **0.82** | **0.87** | **0.72** | **0.74** | **0.70** | **0.74** | **0.70** | **0.68** | **0.68** | **0.72** | **0.73** | 0.76 | **0.70** | 0.75 |
| | GlobalNE | 0.89 | | | | 0.74 | | | | 0.69 | | | | 0.78 | | | |

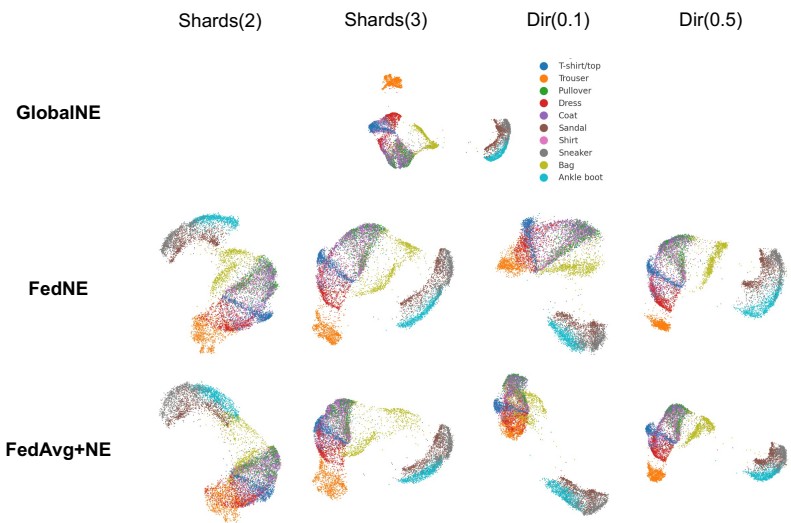

Figure 3: Visualization of the resulting global test 2D embeddings on the Fashion-MNIST dataset under four FL setups of 20 clients ($M = 20$).

Table 3: Ablation study on removing one of the components from FEDNE pipeline with MNIST and scRNA-Seq datasets

| Metric | FedAvg+NE MNIST | RNA-Seq | w/o Data Mixing MNIST | RNA-Seq | w/o Surrogate MNIST | RNA-Seq | FEDNE MNIST | RNA-Seq |
|---|---|---|---|---|---|---|---|---|
| Cont. | 0.97 | 0.97 | 0.96 | 0.97 | 0.96 | 0.97 | 0.97 | 0.97 |
| Trust. | 0.78 | 0.85 | 0.85 | 0.86 | 0.80 | 0.86 | 0.85 | 0.87 |
| $k$NN | 0.48 | 0.85 | 0.65 | 0.87 | 0.54 | 0.88 | 0.72 | 0.90 |
| Stead. | 0.54 | 0.64 | 0.63 | 0.71 | 0.54 | 0.65 | 0.63 | 0.73 |
| Cohes. | 0.77 | 0.68 | 0.84 | 0.71 | 0.81 | 0.69 | 0.82 | 0.70 |

# 6 Discussion

**Privacy Preserving**. As introduced in section 4, FEDNE incorporates the proposed surrogate models into the traditional FEDAVG framework where the surrogate models only take the *low-dimensional* randomly sampled data as inputs. After training, each surrogate model contains much-compressed information about the corresponding client. Thus, we consider the privacy concerns to be alleviated as one cannot directly reconstruct the original high-dimensional client data. To further enhance privacy protection, our framework can be integrated with various privacy-preserving techniques at different stages. For example, Gaussian mechanisms (GM) can be applied to the parameters of the surrogate model before it is sent to the server.

**Scalability and Computational Complexity.** To our knowledge, the field of dimensionality reduction (DR) focuses on relatively smaller-scale datasets, compared to the studies of classification problems. This is because computational complexity is never a trivial problem even for many outstanding DR techniques, particularly for non-linear methods such as Isomap and t-SNE which have non-convex cost functions [44]. In our experiments, we have included the most widely used benchmarks in the DR literature. Moreover, we have considered more participants and larger scales of data compared to prior work [37, 38]. While, unlike the other FL studies focused on classification, our experiments have not yet included much larger datasets or with increased numbers of clients, we expect our approach to be applicable in real-world settings, for example, cross-silo settings with manageable amounts of clients. In terms of computation, as discussed in section 4, our approach requires only 35% additional GPU time compared to FEDAVG, and we expect such overhead to remain similar when going to larger datasets. When the client number increases, we may optionally drop a portion of surrogate models in local training.

# 7 Conclusion

In this paper, we develop a federated neighbor embedding technique built upon the FEDAVG framework, which allows collaboratively training a data projection model without any data sharing. To tackle the unique challenges introduced by the pairwise training objective in the NE problem, we propose to learn a surrogate model within each client to compensate for the missing repulsive forces. Moreover, we conduct local data augmentation via an intra-client data mixing technique to address the incorrect neighborhood connection within a client. We compare FEDNE to four baseline methods and the experiments have demonstrated its effectiveness in preserving the neighborhood data structures and clustering structures.

## ACKNOWLEDGMENTS

We would like to thank the reviewers and Xinyu Zhou for their valuable feedback. This work is supported in part by the NSF-funded AI institute Grant OAC-2112606 and Cisco Research.

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

# Appendix

## A  Experimental Details

### A.1  Local training

In the experiments of MNIST and Fashion-MNIST datasets, we use Adam optimizer with a learning rate of $0.001$ and a batch size of $512$ (i.e., the number of edges in the $k$NN graphs not the number of data instances). The learning rate was decreased by $0.1$ at $30\%$ and $60\%$ of the total rounds. For the experiments with the single-cell RNA-Seq and CIFAR-10 dataset, the learning rate was initially set to $1 \times 10^{-4}$. For negative sampling, we fix the number of non-neighboring data points sampled per edge to be $5$ ($b = 5$).

Moreover, the surrogate loss function is introduced into the local training at $30\%$ of the total rounds, primarily due to the following concerns: during each round of local training, the surrogate loss function in use was constructed using the global NE model from the previous round. Thus, to avoid dramatic deviations between the surrogate function in use and the NE model newly updated by local training, the surrogate function is integrated after the model has already gained a foundational understanding of the data structures and thereby the optimization process tends to be more stable.

### A.2  Surrogate loss function training

The surrogate loss function of each client is fine-tuned at every round from the previous model but the training set (i.e., $D_q = \{(z_{q_i}, l_{q_i}^{rep})\}_{i=1}^{|D_q|}$ in the main paper) needs to be rebuilt according to the current global NE model. The surrogate function is optimized by minimizing the mean squared error (MSE) using the Adam optimizer with a learning rate of $0.001$.

## B  Design Choices and Hyperparameter Selections

### B.1  Choice of $k$ in building local $k$NN graphs

In the main experiments, a fixed number of neighbors ($k$=7) is used for building clients' $k$NN graphs. We conduct further experiments using different $k$ values under the setting of $Dirichlet(0.1)$ on the MNIST dataset with 20 clients. The results are shown in Table 4 We found that within a certain range (i.e., 7 to 30), the performance of FedNE is relatively stable. When k is too large (e.g., k=50), the performance drops but our FedNE still outperforms the baseline methods, FedAvg+NE. Overall, this trend aligns with the general understanding of dimensionality reduction methods.

Table 4: Experiments on different numbers of neighbors $k$ in building the local $k$NN graphs. The experiments are conducted under the setting of $Dirichlet(0.1)$ on the MNIST dataset with 20 clients. We found that within a certain range (i.e., 7 to 30), the performance of FedNE is relatively stable.

| Metric | Method | $k = 7$ | $k = 15$ | $k = 30$ | $k = 50$ |
|--------|--------|---------|----------|----------|----------|
| Conti. | FedAvg+NE | **0.97** | 0.96 | **0.96** | **0.96** |
|        | FEDNE | **0.97** | **0.97** | 0.96 | 0.96 |
| Trust. | FedAvg+NE | 0.78 | 0.77 | 0.77 | 0.77 |
|        | FEDNE | **0.85** | **0.85** | **0.84** | **0.79** |
| $k$NN  | FedAvg+NE | 0.48 | 0.47 | 0.45 | 0.45 |
|        | FEDNE | **0.72** | **0.69** | **0.67** | **0.54** |
| Stead. | FedAvg+NE | 0.54 | 0.50 | 0.51 | 0.51 |
|        | FEDNE | **0.63** | **0.64** | **0.62** | **0.55** |
| Cohes. | FedAvg+NE | 0.77 | 0.75 | 0.77 | **0.77** |
|        | FEDNE | **0.82** | **0.84** | **0.82** | **0.77** |

### B.2  Weights in intra-client data mixing strategy

We fixed the $\alpha$ to be $0.2$ in the main experiments to perform intra-client data augmentation. Here, we alter the weight used in the intra-client data mixing strategy. We adjust the mixing weight $\lambda$ by changing the $\alpha$ value, where $\lambda \sim \text{Beta}(\alpha)$. The experiments are conducted under the setting of $Dirichlet(0.1)$ on the MNIST dataset with 20 clients. In the ablation study shown in Section 5.3 Table 3, we demonstrated the effectiveness of adding our intra-client data mixing strategy. These

Figure 4: Experimental results on different step sizes in grid sampling for training the surrogate models. The experiments are conducted under the setting of $Dirichlet(0.1)$ on the MNIST dataset with 20 clients. In the main paper, the default step size is set to $0.3$. The results demonstrate that the performance of FedNE is stable when the step size is below $1.0$.

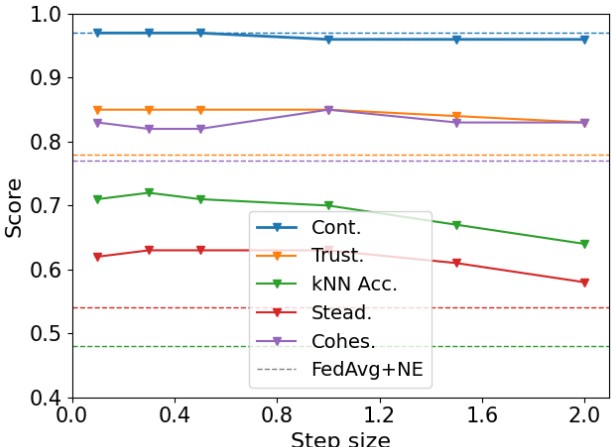

additional results shown in Table 5 demonstrate that FedNE is very stable across different mixing weights.

Table 5: Experimental results on different weights used in intra-client data mixing strategy. We adjust the mixing weight $\lambda$ by changing the $\alpha$ value, where $\lambda \sim \text{Beta}(\alpha)$. The experiments are conducted under the setting of $Dirichlet(0.1)$ on the MNIST dataset with 20 clients. These additional results demonstrate that FedNE is very stable across different mixing weights.

| Metric | Method | $\alpha = 0.1$ | $\alpha = 0.2$ | $\alpha = 0.3$ | $\alpha = 0.4$ |
|---|---|---|---|---|---|
| Conti. | FedAvg+NE | | | 0.97 | |
| | FEDNE | 0.97 | 0.97 | 0.97 | 0.97 |
| Trust. | FedAvg+NE | | | 0.78 | |
| | FEDNE | 0.85 | 0.85 | 0.85 | 0.85 |
| $k$NN | FedAvg+NE | | | 0.48 | |
| | FEDNE | 0.72 | 0.72 | 0.72 | 0.72 |
| Stead. | FedAvg+NE | | | 0.54 | |
| | FEDNE | 0.63 | 0.63 | 0.62 | 0.63 |
| Cohes. | FedAvg+NE | | | 0.77 | |
| | FEDNE | 0.83 | 0.82 | 0.83 | 0.84 |

### B.3 Choice of step size for grid sampling

The step size is used to control the resolution of grid sampling for training the surrogate models. In the main paper, the default step size is set to $0.3$, and here, we experiment with using different step sizes. The experiments are conducted under the setting of $Dirichlet(0.1)$ on the MNIST dataset with 20 clients. The results in Figure 4 demonstrate that the performance of FedNE is stable when the step size is below $1.0$. However, when we increase the step size beyond $1.0$, we observe a gradual decrease in performance, especially in terms of kNN classification accuracy and steadiness metrics. Despite this, FedNE still maintains better performance than FedAvg+NE.

### B.4 Data source for training surrogate models

To construct the training set of the surrogate loss function in a more comprehensive and manageable way, each client employs a grid-sampling strategy in the 2D space. Here, we conduct experiments on MNIST and Fashion-MNIST datasets to compare the performance between using our grid-sampling strategy and using only local 2D embeddings as the training data. Table 6 and Table 7 show the comparison results for MNIST and Fashion-MNIST test data, respectively. We highlight the better

results in both tables. The grid-sampling method outperforms the baseline approach (i.e., only using local embedding in surrogate function training), while the baseline still achieves better performance compared to FedAvg+NE. Overall, the results validate the effectiveness of employing the surrogate loss function during local training and also support our proposed grid-sampling strategy.

Table 6: Quantitative comparison between the performance of using our grid-sampling strategy and using only local 2D embeddings as surrogate training data. The following results are experimented with the FL setting of 20 clients and two different Shards partitions on the **MNIST** test data.

| Metric | Method | Shards(2) | | Shards(3) | |
|---|---|---|---|---|---|
| | | Local | Grid | Local | Grid |
| Cont. | GlobalNE | 0.97 | | 0.97 | |
| | FEDNE | **0.96** | **0.96** | 0.96 | **0.97** |
| | FedAvg+NE | 0.96 | | 0.97 | |
| Trust. | GlobalNE | 0.94 | | 0.94 | |
| | FEDNE | 0.86 | **0.87** | 0.90 | **0.91** |
| | FedAvg+NE | 0.83 | | 0.89 | |
| $k$NN | GlobalNE | 0.93 | | 0.93 | |
| | FEDNE | 0.69 | **0.71** | 0.84 | **0.87** |
| | FedAvg+NE | 0.54 | | 0.73 | |

Table 7: Quantitative comparison between the performance of using our grid-sampling strategy and using only local 2D embeddings as surrogate training data. The following results are experimented with the FL setting of 20 clients and two different Shards partitions on the **Fashion-MNIST** test data.

| Metric | Method | Shards(2) | | Shards(3) | |
|---|---|---|---|---|---|
| | | Local | Grid | Local | Grid |
| Cont. | GlobalNE | 0.99 | | 0.99 | |
| | FEDNE | 0.98 | **0.99** | 0.98 | **0.99** |
| | FedAvg+NE | 0.99 | | 0.99 | |
| Trust. | GlobalNE | 0.97 | | 0.97 | |
| | FEDNE | 0.95 | **0.96** | 0.93 | **0.96** |
| | FedAvg+NE | 0.93 | | 0.96 | |
| $k$NN | GlobalNE | 0.74 | | 0.74 | |
| | FEDNE | 0.65 | **0.67** | 0.64 | **0.69** |
| | FedAvg+NE | 0.59 | | 0.66 | |

## B.5 Frequency of surrogate function update

In all experiments, the surrogate loss functions are retrained at every round. While frequent retraining introduces computational burdens for each client, using outdated surrogate loss functions can bias the optimization process of the repulsive loss. Thus, we conduct experiments on the MNIST dataset to showcase the impacts of the frequency of surrogate function updates. We conducted experiments with the other four setups, i.e., retraining the surrogate function every 5, 10, 15, or 20 more rounds. The default one is updating the surrogate function at every round. According to the line chart in Figure 5, the performance decreased dramatically only when the surrogate loss function was updated more than every 10 rounds.

## B.6 Time to integrate the surrogate loss function

In the main experiments, the surrogate loss function is integrated into local training after $30\%$ of the total rounds have been finished. Here, we conduct further experiments on introducing the surrogate function at different time periods to confirm our decision, and the results are demonstrated in Figure 6. First, the continuity is not affected too much and retains high scores under various setups. However, when the surrogate loss function is introduced too early, the trustworthiness and $k$NN accuracy drops, which may indicate that the inter-client repulsion is better to be involved after the intra-client forces have become relatively stable. Moreover, the performance of $55\%$ also drops, particularly on $k$NN accuracy. This could be because the training process of FEDAVG has converged, making it too late to integrate additional constraints into the training procedure.

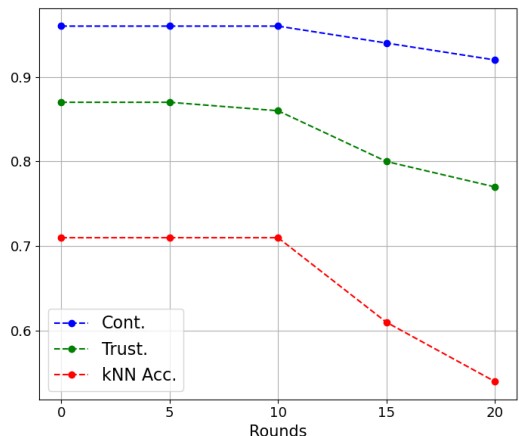

Figure 5: The quantitative evaluation on testing the surrogate function under five different retraining frequencies. The line chart shows the results of the MNIST test data with the FL setting of 20 clients and the Shards partition with two classes of data per client.

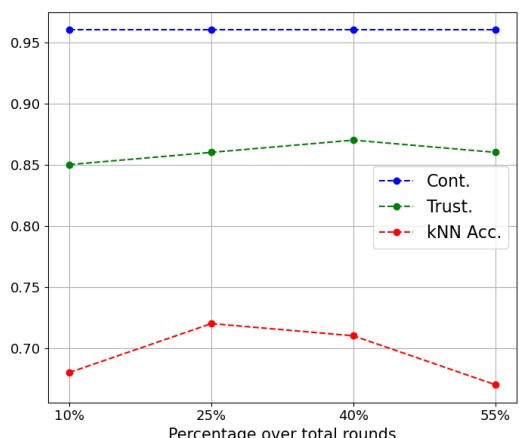

Figure 6: The quantitative evaluation on four different periods to introduce the surrogate function. The line chart shows the comparison results on the MNIST test data with the FL setting of 20 clients and the Shards partition with two classes of data per client.

## C  Partial Client Participation

We simulate the straggler problem by randomly sampling 10% of the clients involved in each round of communication under the setting of $Dirichlet(0.1)$ on the MNIST dataset with 100 clients. While the performance under partial client participation is worse than under full client participation, the results in Table 8

show that FEDNE still performs notably better than the baseline method, FedAvg+NE.

## D  Evaluation Results on the Shards Setting

In addition to the table in the main paper, we also report the results of the Shards setting on MNIST, Fashion-MNIST, and CIFAR-10 datasets in Table 9.

## E  Visualization Results by FEDNE

We demonstrate the comprehensive visualization results from the centralized setting i.e., GlobalNE, our FEDNE and FedAvg+NE, and FedProx+NE on MNIST, Fashion-MNIST, RNA-Seq, and CIFAR-10 global test data. The five metrics are introduced in the main paper.

Table 8: We experiment with partial client participation by randomly sampling 10% of the clients involved in each communication round under the setting of $Dirichlet(0.1)$ on the MNIST dataset with 100 clients. While the performance under partial client participation is worse than under full participation, the results of FEDNE are still notably better than the baseline method.

| Metric | Method | 100% Participation | 10% Participation |
|---|---|---|---|
| Conti. | FedAvg+NE | **0.96** | 0.95 |
| | FEDNE | **0.96** | **0.96** |
| Trust. | FedAvg+NE | 0.74 | 0.73 |
| | FEDNE | **0.82** | **0.79** |
| $k$NN | FedAvg+NE | 0.43 | 0.41 |
| | FEDNE | **0.65** | **0.58** |
| Stead. | FedAvg+NE | 0.43 | 0.41 |
| | FEDNE | **0.57** | **0.50** |
| Cohes. | FedAvg+NE | 0.74 | 0.72 |
| | FEDNE | **0.82** | **0.80** |

Table 9: Quality of the resulting 2D embedding under the Shards setting ($C = 2$ and $C = 3$) on global test data of the tree datasets. FEDNE achieves top performance on preserving both neighborhood structures (i.e., continuity, trustworthiness, and $k$NN classification accuracy) and global inter-cluster structures (i.e., steadiness and cohesiveness).

| | | MNIST | | | | Fashion-MNIST | | | | CIFAR-10 | | | |
|---|---|---|---|---|---|---|---|---|---|---|---|---|---|
| | | $M = 20$ | | $M = 100$ | | $M = 20$ | | $M = 100$ | | $M = 20$ | | $M = 100$ | |
| Metric | Method | $C=2$ | $C=3$ | $C=2$ | $C=3$ | $C=2$ | $C=3$ | $C=2$ | $C=3$ | $C=2$ | $C=3$ | $C=2$ | $C=3$ |
| Conti. | LocalNE | 0.90 | 0.92 | 0.93 | 0.94 | 0.96 | 0.97 | 0.97 | 0.97 | 0.85 | 0.89 | 0.87 | 0.89 |
| | FedAvg+NE | 0.96 | 0.97 | 0.97 | 0.97 | 0.99 | 0.99 | 0.99 | 0.99 | 0.93 | 0.94 | 0.93 | 0.94 |
| | FedProx+NE | 0.97 | 0.97 | 0.97 | 0.97 | 0.99 | 0.99 | 0.98 | 0.99 | 0.93 | 0.94 | 0.93 | 0.94 |
| | **FedNE** | 0.96 | 0.97 | 0.97 | 0.97 | 0.99 | 0.99 | 0.99 | 0.99 | 0.93 | 0.94 | 0.93 | 0.94 |
| | GlobalNE | | | 0.97 | | | | 0.99 | | | | 0.95 | |
| Trust. | LocalNE | 0.74 | 0.77 | 0.75 | 0.78 | 0.88 | 0.91 | 0.90 | 0.92 | 0.72 | 0.77 | 0.72 | 0.76 |
| | FedAvg+NE | 0.83 | 0.89 | 0.84 | 0.89 | 0.93 | 0.96 | 0.94 | 0.96 | 0.80 | 0.85 | 0.80 | 0.83 |
| | FedProx+NE | 0.82 | 0.88 | 0.84 | 0.89 | 0.94 | 0.96 | 0.93 | 0.96 | 0.80 | 0.85 | 0.80 | 0.83 |
| | **FedNE** | 0.87 | 0.91 | 0.87 | 0.90 | 0.96 | 0.96 | 0.95 | 0.96 | 0.81 | 0.85 | 0.82 | 0.84 |
| | GlobalNE | | | 0.94 | | | | 0.97 | | | | 0.87 | |
| $k$NN | LocalNE | 0.41 | 0.47 | 0.41 | 0.50 | 0.53 | 0.56 | 0.52 | 0.57 | 0.40 | 0.47 | 0.40 | 0.48 |
| | FedAvg+NE | 0.54 | 0.73 | 0.54 | 0.73 | 0.59 | 0.66 | 0.60 | 0.65 | 0.51 | 0.64 | 0.533 | 0.65 |
| | FedProx+NE | 0.55 | 0.73 | 0.55 | 0.73 | 0.59 | 0.66 | 0.60 | 0.65 | 0.52 | 0.65 | 0.524 | 0.65 |
| | **FedNE** | 0.71 | 0.87 | 0.73 | 0.80 | 0.67 | 0.69 | 0.66 | 0.67 | 0.56 | 0.70 | 0.582 | 0.72 |
| | GlobalNE | | | 0.93 | | | | 0.73 | | | | 0.78 | |
| Stead. | LocalNE | 0.45 | 0.47 | 0.46 | 0.51 | 0.61 | 0.68 | 0.64 | 0.70 | 0.54 | 0.59 | 0.55 | 0.59 |
| | FedAvg+NE | 0.53 | 0.66 | 0.52 | 0.68 | 0.75 | 0.83 | 0.70 | 0.82 | 0.63 | 0.69 | 0.66 | 0.69 |
| | FedProx+NE | 0.52 | 0.60 | 0.55 | 0.67 | 0.75 | 0.81 | 0.74 | 0.81 | 0.63 | 0.70 | 0.66 | 0.69 |
| | **FedNE** | 0.64 | 0.72 | 0.64 | 0.71 | 0.81 | 0.82 | 0.80 | 0.82 | 0.66 | 0.71 | 0.67 | 0.70 |
| | GlobalNE | | | 0.76 | | | | 0.83 | | | | 0.73 | |
| Cohes. | LocalNE | 0.69 | 0.74 | 0.70 | 0.74 | 0.61 | 0.65 | 0.63 | 0.66 | 0.63 | 0.67 | 0.63 | 0.67 |
| | FedAvg+NE | 0.81 | 0.86 | 0.82 | 0.88 | 0.68 | 0.72 | 0.71 | 0.72 | 0.71 | 0.75 | 0.72 | 0.74 |
| | FedProx+NE | 0.81 | 0.86 | 0.83 | 0.88 | 0.69 | 0.73 | 0.69 | 0.72 | 0.71 | 0.76 | 0.71 | 0.75 |
| | **FedNE** | 0.86 | 0.88 | 0.85 | 0.88 | 0.70 | 0.73 | 0.71 | 0.73 | 0.72 | 0.73 | 0.71 | 0.75 |
| | GlobalNE | | | 0.89 | | | | 0.74 | | | | 0.78 | |

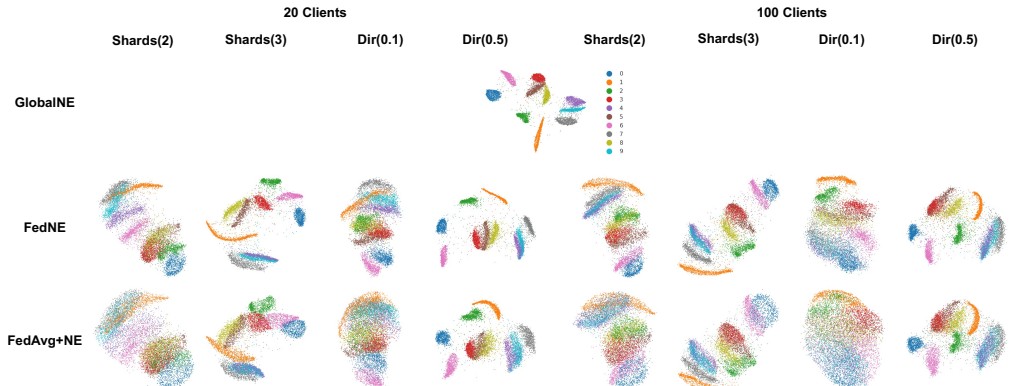

Figure 7: Visualization results from centralized setting, FEDNE and FEDAVG on **MNIST** test dataset under eight different FL settings (i.e., Shards with two classes or three classes per client; and Dirichlet with $\alpha = 0.1$ or $\alpha = 0.5$).

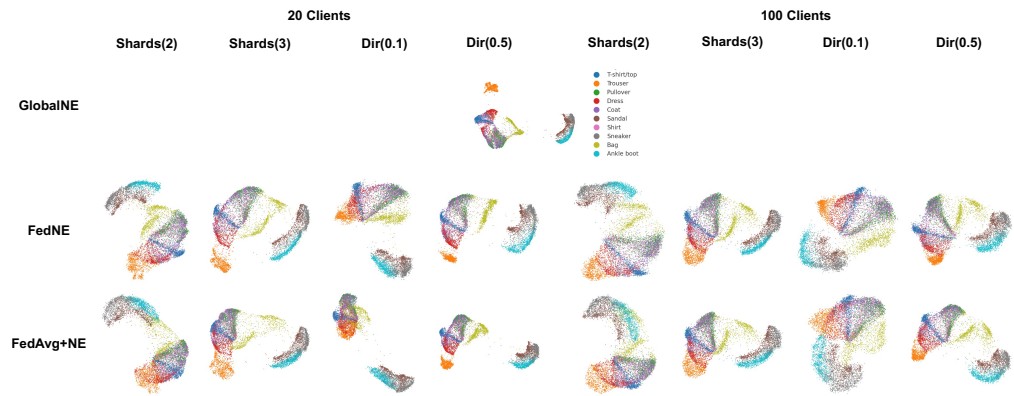

Figure 8: Visualization results from centralized setting, FEDNE and FEDAVG on **Fashion-MNIST** test dataset under eight different FL settings (i.e., Shards with two classes or three classes per client; and Dirichlet with $\alpha = 0.1$ or $\alpha = 0.5$).

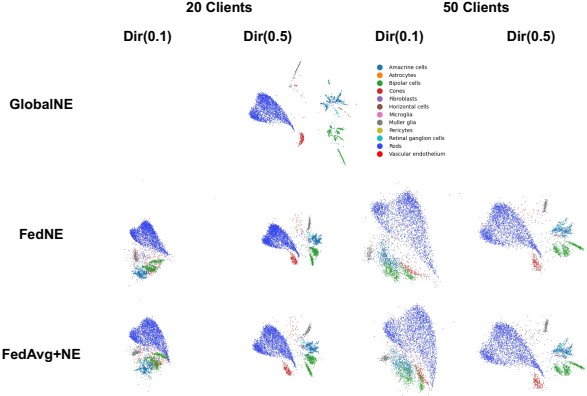

Figure 9: Visualization results from centralized setting, FEDNE and FEDAVG on **scRNA-Seq** test dataset under four different FL settings (i.e., Dirichlet with $\alpha = 0.1$ or $\alpha = 0.5$).

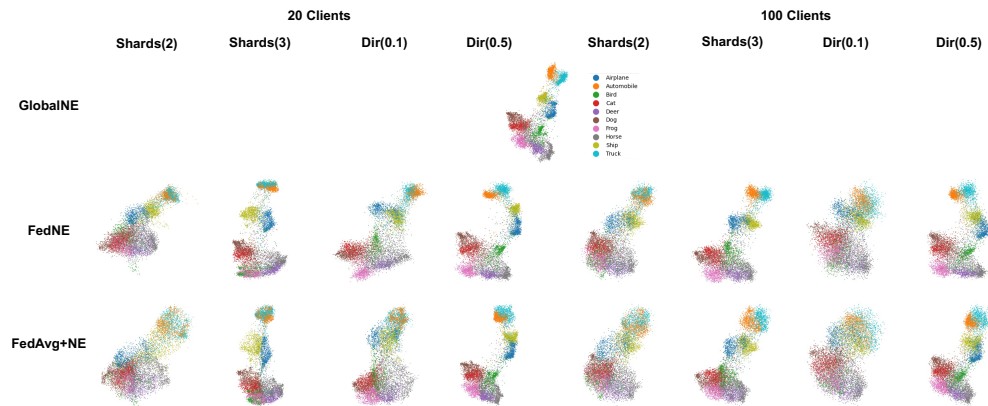

Figure 10: Visualization results from centralized setting, FEDNE and FEDAVG on **scRNA-Seq** test dataset under four different FL settings (i.e., Dirichlet with $\alpha = 0.1$ or $\alpha = 0.5$).

