# OpenReview forum: "FedNE: Surrogate-Assisted Federated Neighbor Embedding for Dimensionality Reduction"
_NeurIPS.cc/2024/Conference — NeurIPS 2024 poster_

### Official Review · Reviewer_y2ZN · 2024-07-06

**Soundness:** 2
**Presentation:** 2
**Contribution:** 3
**Rating:** 5
**Confidence:** 4

**Summary:**

The paper introduces a novel approach to address the challenge of collaboratively visualizing high-dimensional data in a federated learning (FL) environment. The proposed method, FEDNE, integrates the FEDAVG framework with contrastive neighbor embedding (NE) techniques, aiming to preserve data privacy while ensuring effective data visualization. By employing a surrogate loss function and an intra-client data mixing strategy, FEDNE seeks to enhance the alignment and preservation of neighborhood structures in the global embedding space. The paper includes comprehensive experiments on both synthetic and real-world datasets, demonstrating the effectiveness of FEDNE in outperforming several baseline methods in terms of neighborhood data structure preservation and clustering.

**Strengths:**

1. FEDNE introduces a novel integration of FEDAVG with contrastive NE techniques, addressing the unique challenges of pairwise data relationships in federated learning environments without requiring data sharing.
2. The intra-client data mixing strategy effectively enhances local data diversity, mitigating the limitations of biased local kNN graphs and ensuring better neighborhood representation.
3. The paper provides a thorough evaluation of FEDNE using various datasets and metrics, showcasing its superior performance compared to baseline methods in preserving neighborhood structures and clustering.

**Weaknesses:**

1.	While the authors mention that FEDNE introduces only 35% more GPU time compared to FEDAVG, the overall complexity and scalability in a more extensive, real-world setting are not fully addressed. The authors should further investigate how FEDNE scales with a significantly larger number of clients and more complex datasets or models.
2.	The paper proposes intra-client data mixing as a solution to the bias in local kNN graphs. However, this approach might not entirely mitigate the issue of incorrect neighbor connections, especially in highly imbalanced datasets. More detailed comparisons with alternative methods or further enhancements could provide a more robust solution.
3.	The focus is primarily on dimensionality reduction. The validation results are performed only on the vision classification tasks. Extending the discussions and analyses to include applications in other domains could be beneficial.

**Questions:**

1.	Could you provide more details on the process of training the surrogate models? Specifically, how do you ensure that these models effectively capture the repulsive forces between dissimilar data points across different clients?
2.	Non-IID data is a common challenge in federated learning. How does FEDNE handle extreme cases of non-IID data distribution? Have you considered any additional mechanisms to ensure robustness in such scenarios?
3.	How sensitive is FEDNE to the choice of hyperparameters, such as the step size for grid sampling, the number of neighbors in kNN, and the weight in intra-client data mixing? Have you performed any sensitivity analysis?

**Limitations:**

The authors have addressed the works limitations and social impacts.

---

> ### Author Rebuttal · Authors · 2024-08-07
>
> > Questions regarding scalability and complexity
>
> Please see the general response.
>
> > The paper proposes intra-client data mixing … However, this approach might not entirely mitigate the issue … More detailed comparisons with alternative methods …
>
> Thank you for the insightful comment. As we pointed out in the paper, the bias in the local kNN graphs is one key blocker to the problem of Federated NE. Dealing with attraction term is more challenging than repulsion term as it requires knowing true data affinity across different clients (as discussed in Sect. 4.2). Such a problem has never been pointed out in prior work, but as emphasized in our paper (Line 209-217).
>
> With this in mind, we made one of the first attempts to mitigate it. We acknowledge that our approach may not fully address the problem, but it demonstrates that reducing its effect can notably improve the overall performance (see Sect.5.3), setting up the foundation for future work to inspire more research in this new field including but not limited to addressing the attractive loss within the settings of FL.
>
> We appreciate your comments on “alternative methods.” To our knowledge, none of the prior work in FL for dimensionality reduction aimed to address this issue. We would appreciate if you could provide references to the alternative methods you mentioned, and we will be happy to provide more discussions in the reviewer/author discussion phase.
>
>
> > … The validation results are performed only on the vision classification tasks … other domains could be beneficial.
>
> Thanks for your valuable comment. We totally agree that the evaluation of our method is mainly on the vision datasets. The main reason we designed the experimental study in this way is mainly to follow a similar paradigm as other DR studies [33,44,55], with the purpose of fair comparisons. However, we admit that only evaluating vision data might be not enough. Therefore, in addition to the commonly used benchmark vision data, we also evaluated our method on **a biological dataset, scRNA-Seq** (see Tables 1 and 2 in the original manuscript). The scRNA-Seq dataset is a real-world dataset containing a collection of gene expression profiles obtained from individual cells of the mouse retina. We hope this additional biological dataset can give you a more comprehensive understanding of our effectiveness in different domains.
>
> [33] “UMAP: Uniform manifold approximation and projection for dimension reduction.” arXiv preprint arXiv:1802.03426, 2018.
>
> [44] "Visualizing data using t-SNE." JMLR 9.11 (2008).
>
> [55] "SpaceMAP: Visualizing High-Dimensional Data by Space Expansion." ICML. 2022.
>
> > Could you provide more details on the process of training the surrogate models? Specifically, how do you ensure that these models effectively capture the repulsive forces between dissimilar data points across different clients?
>
> Thanks for the comment. We have discussed in Sect. 4.1 and A.2 in the appendix, but we should have made this more clear. To train the surrogate repulsion model for a client $m$, we generate a set of low-dimensional (low-D) query points via grid sampling to serve as potential embedding positions of other clients’ data points. Then, for each newly sampled low-D query data $z_q$, we pre-compute the repulsive loss between $z_q$ and $b$ data points sampled within client $m$, which serve as the training targets.
>
> The surrogate model is expected to learn a mapping from these newly sampled embedding positions to their corresponding repulsion loss, as measured within client $m$. According to the repulsive loss term introduced in Section 3.1, a larger low-D Euclidean distance will result in a smaller repulsive loss value [6]. Thus, the repulsion loss decreases as the distance between the data pairs increases. If two embedding points are close within a threshold, our surrogate model is able to approximate the repulsion force; if the points are far apart, the model estimates the repulsion force between them to be close to zero.
>
> [6] "Attraction-repulsion spectrum in neighbor embeddings." JMLR 23.95 (2022): 1-32.
>
> > How does FEDNE handle extreme cases of non-IID data distribution? Have you considered any additional mechanisms to ensure robustness in such scenarios?
>
> We agree that the ability to handle non-IID data is very important. Therefore, we have conducted experiments on the settings of Dirichlet(0.1) and Shards with C=2 to demonstrate the effectiveness of FedNE in handling extreme non-IID cases, as shown in Tables 2 and 6 in the original manuscript.
>
> FedNE **addresses non-IID data distribution *exactly* via the surrogate models**. For example, under an *extreme* non-IID condition, each client may have data exclusively for certain clusters. In this scenario, the issue with the attraction term might be *relaxed* because the true neighboring points are likely to reside within the same client. However, in the global context, the dissimilar data pairs contributing to the repulsive loss term are located across different clients. As discussed in Sect. 3.3 and 4.1, our surrogate model is specifically designed for this condition by filling in the missing repulsion terms.
>
> > How sensitive is FEDNE to the choice of hyperparameters
>
> Thanks for the valuable suggestions. We have included experiments in Appendix B to analyze the sensitivity of some important hyperparameters in our method such as the frequency of surrogate function updates and the time to integrate the surrogate loss function into the local training.
>
> Furthermore, we want to provide additional results on analyzing the sensitivity for the following hyperparameters: (1) Number of neighbors in the kNN graphs, (2) Step sizes in the grid sampling, and (3) The weight in intra-client data mixing. Due to the space limit, we included the detailed analysis in the **figure captions**. **Please find the results and analysis in the attached PDF**.

---

> > ### Comment · Reviewer_y2ZN · 2024-08-12
> >
> > Thank you for your reply. I think the authors have addressed my concerns, especially with regard to the scalability and generality of the proposed method. Therefore, I decide to change the score to borderline accept.

---

> ### Author Response · Authors · 2024-08-12
> **Re: Official Comment by Reviewer y2ZN**
>
> Thank you for your feedback and your willingness to increase the rating. We will incorporate our rebuttal into the revised version of our paper, and we would appreciate it if you would be supportive of the acceptance of our paper.

---

### Official Review · Reviewer_mP1Q · 2024-07-10

**Soundness:** 2
**Presentation:** 3
**Contribution:** 2
**Rating:** 3
**Confidence:** 5

**Summary:**

The paper "FEDNE: Surrogate-Assisted Federated Neighbor Embedding for Privacy-Preserving Dimensionality Reduction" presents a method for visualizing high-dimensional data while maintaining privacy without requiring any shareable reference data.

Federated Neighbor Embedding (FEDNE): A framework combining federated averaging (FEDAVG) with contrastive neighbor embedding (NE) to create a joint NE model across multiple clients without compromising data privacy.

Surrogate Loss Function: An innovative loss function to enhance inter-client repulsion in the global embedding space, ensuring better separation of data points from different clients while preserving local data structures.

Data-Mixing Strategy: A technique to counter issues like invisible and false neighbors in local k-nearest neighbor (kNN) graphs by mixing data from various clients during training, thus improving the quality of the learned embeddings.

**Strengths:**

Well-Presented: The paper is clearly and coherently written, making it easy to follow.
Novel Approach: The study addresses an important problem with a novel approach, combining federated learning with neighbor embedding techniques.

**Weaknesses:**

Privacy Concerns: While the approach is innovative, the paper does not sufficiently address privacy concerns. It lacks experiments and guarantees demonstrating the privacy preservation of the FedNE approach.

Computational Inefficiency: The method appears to be computationally inefficient. There are no experiments conducted on large datasets, such as those in real-world medical or other privacy-critical domains, where computational complexity could be a significant issue.

Inadequate Analysis of Related Work: The related works section is not thoroughly analyzed or discussed, missing critical comparisons and context necessary for a comprehensive understanding of the state of the art.

The study's applicability could be strengthened by extending beyond benchmark datasets to encompass real-world, privacy-sensitive datasets found in domains such as healthcare or finance. This expansion would provide a more robust demonstration of the method's practical relevance and effectiveness.

Additionally, addressing pairwise issues associated with attraction terms is essential for improving the preservation of neighborhood structures and enhancing clustering quality.

Furthermore, it is crucial to conduct thorough analyses aimed at optimizing the computational efficiency and scalability of the algorithms, ensuring their capability to handle large-scale datasets effectively. Moreover, the method currently lacks explicit consideration of privacy guarantees. And on elucidating how privacy concerns are addressed within the framework and formalizing privacy guarantees to assure users and stakeholders.

**Questions:**

Privacy Guarantees: The paper lacks a thorough discussion on the privacy guarantees of the proposed method, especially against adversarial attackers.The experimental evaluation focuses solely on utility results, with no evaluation of data privacy. How is privacy preservation quantified and ensured? What is the acceptable level of privacy preservation? The paper should include theoretical arguments and experiments demonstrating actual privacy management. From a privacy perspective, it would be helpful to provide guidance on the limitations of this method, particularly regarding transparency and explainability (e.g., OECD AI principle 1.3). What measures are in place to address these concerns?

Experiments: No experiments are conducted on downstream tasks related to the problem, aside from analyzing structural properties? Moreover,  no experiments conducted on large datasets? Lastly, how do the authors plan to tackle the problem of heavy computation? The method appears computationally intensive, which could hinder its practicality.

Reconstruction from Gradients: According to Zhu and Han’s study [1], model gradients can be used in some scenarios to partially reconstruct client data. How does the proposed method address this issue?

The paper claims to operate without relying on shareable reference data, yet it utilizes additional 2D data points from grid sampling to estimate the training targets via repulsion loss and additional augmented data points via interpolation for the attractive loss. Given this, how does the strategy address the significant computational burden it introduces, and is it feasible for real-world applications where computational efficiency is critical?

[1] Ligeng Zhu, Zhijian Liu, and Song Han. 2019. Deep leakage from gradients. Advances in neural information processing systems 32 (2019)

**Limitations:**

No societal impacts.

---

> ### Author Rebuttal · Authors · 2024-08-07
>
> > Questions related to privacy concerns
>
> We found that major concerns mentioned in the weakness and questions are related to “privacy-preserving”, and these concerns may arise from the “privacy-preserving” term in our paper title. First, we want to apologize for the confusion caused by our title. The main focus of our paper is on a FL setting, aiming to bring up the unique FL challenges and propose effective solutions for the pairwise training objective in the problem of Federated Neighbor Embedding. Besides the privacy-preserving properties that a Federated setting has introduced, we do not specifically develop any privacy-preserving mechanisms and we certainly do not claim that our approach introduces additional privacy guarantees. **We will surely refine our title and any writing to clarify this**.
>
> Introducing formal privacy guarantees is not a trivial task. Among the only two existing methods, dSNE [37] proposes a decentralized neighbor embedding framework and later, they extend dSNE to be privacy-guaranteed by incorporating differential privacy (DP) as F-dSNE [38]. We have to admit that, developing privacy-guaranteed Federated NE methods while addressing the lack of inter-client data relationships is an important but challenging task. Therefore, **we will consider the privacy of FedNE as an orthogonal problem and will work toward incorporating DP or other techniques in our future work**. We will add discussions on how to enhance FedNE with privacy-preserving techniques.
>
> > Inadequate Analysis of Related Work
>
> Thanks for the valuable comment. As our work intersects with two research areas: FL and DR, we have structured the related work section into three subtopics in Sect. 2. In fact, we have compared FedNE with two closely related works, dSNE and F-dSNE in Sect. 2. However, due to the page limitations, we might have overlooked some relevant literature and we are happy to include additional work on FL and DR. However, we kindly believe that our literature review in the field of decentralized DR is thorough.
>
> > The study's applicability could be strengthened … to encompass real-world, privacy-sensitive datasets …
>
> Thanks for the constructive suggestions. As clarified in our above response, our main focus is on FL for DR instead of particularly improving the privacy-preserving properties. Thus, we focus more on benchmark datasets in the field of DR. Specifically, we have included the standard datasets in the DR community but with more practical settings (i.e., more clients and larger datasets) compared to prior work (please see the general response).
>
> With that being said, your comments are well received, and we will consider extending our study to some real-world privacy-sensitive datasets. However, we do want to point out some considerations. Datasets from different domains have unique properties. For example, the prior work [37] mentioned that, in the neuroimaging domain, not all data is private/unshareable, and many public MRI datasets are accessible. The only existing works [37,38] related to our problem focusing on neuroimaging data have made this unreasonable assumption which makes their method hard to generalize. In contrast, to prevent loss of generalizability, our FedNE was developed without assuming any domain-specific properties. Furthermore, we believe that adapting FedNE to the domain-specific datasets should not be difficult as we did not make any prior assumptions about the data properties.
>
> > Questions regarding scalability, complexity, and computational efficiency
>
> Please see the general response.
>
> > The paper claims to operate without relying on shareable reference data, yet it utilizes additional 2D data points from grid sampling to estimate the training targets …
>
> Thanks for your comment. We want to reiterate that our FedNE does not use any shareable reference data, as the publicly available dataset is often inaccessible in many real-world applications (Line 112-114). Moreover, the quality of the reference data can significantly affect the performance [37]. Given these more practical considerations, we propose surrogate models and intra-client data mixing to address the constraints and challenges brought by the Federated setting. We think **the computational cost is still manageable**, and more importantly, **our work is valuable as it was developed within a more practical and general FL setting**, even with some extra burden from data sampling.
>
> > No experiments are conducted on downstream tasks …
>
> Thanks for your suggestion. We want to emphasize that the main focus of this work is to propose a DR framework within the FL setting. Therefore, most of our evaluation metrics are designed for analyzing the structural properties of the embeddings. This is because metrics like preservation of neighborhood relationships and preservation of data clusters can provide a clear indication of how well our DR model performs [57,58] regardless of whether it is under the federated or centralized setting. We totally agree that evaluating on other downstream tasks might be valuable to provide a different understanding of our model performance. However, the metric to evaluate other downstream tasks might not be able to accurately assess our capability of preserving the neighborhood relationship as a DR framework. Thus, we respectfully think conducting experiments on downstream tasks e.g., classification might be biased to assess our effectiveness in DR.
>
> [37] "See without looking: joint visualization of sensitive multi-site datasets." IJCAI. 2017.
>
> [38] "Federated, Fast, and Private Visualization of Decentralized Data." Workshop of Federated Learning and Analytics in Practice (2023).
>
> [57] "Toward a quantitative survey of dimension reduction techniques." IEEE TVCG 27.3 (2019): 2153-2173.
>
> [58] "Dimensionality reduction: A comparative review." JMLR 10.66-71 (2009): 13.
>
> [59] "Feature dimensionality reduction: a review." Complex & Intelligent Systems 8.3 (2022): 2663-2693.

---

> > ### Comment · Reviewer_mP1Q · 2024-08-12
> > **Response to Author's rebuttal**
> >
> > Thank you for your response. However, my concerns about the privacy aspects of the proposed method remain unresolved. Federated learning, by design, aims to decentralize and collaboratively train models in a privacy-preserving manner, which inherently implies a need for privacy. This is precisely why one would choose this approach. However, recent research suggests that federated learning on its own may not be sufficient to guarantee privacy. Therefore, I will maintain my score.

---

> > > ### Author Response · Authors · 2024-08-14
> > >
> > > Dear Reviewer mP1Q,
> > >
> > > We appreciate your timely feedback. We understand your point about the privacy concern of the general Federated Learning (FL) framework. To our understanding, FL was originally conceived to allow model training without centralized data collection. Such a framework enables users to keep their data without sharing. Since then, a significant portion of the work in FL has aimed to improve the performance of FL in terms of its accuracy and convergence, especially under heterogeneous settings and communication constraints. Meanwhile, another branch of the FL research focuses on the privacy guarantee as you mentioned, as they found that decentralizing the data alone is not sufficient to protect privacy. We think both branches (and several others, e.g., system-level consideration of FL) are valuable --- *while one single paper may not be able to address both aspects*, together they aim for a synergy to make FL a robust, widely applicable, and privacy-preserving framework.
> > >
> > > As highlighted in a recent survey [1], FL still faces multiple unresolved challenges or deficiencies, including privacy protection, communication costs, systems heterogeneity, and deepening the research of FL in various fields. In Section 7, the survey paper said, “The development of FL is faced with multiple challenges, and no single strategy can comprehensively solve these bottlenecks in the practical application of FL technology.”
> > >
> > > In our paper, we aim to extend FL's applicability (to a rarely studied ML problem) and address the associated performance challenges. We kindly want to reiterate that our focus is not to improve the general FL framework in terms of its privacy aspect but to explore a novel application domain of FL.
> > >
> > > We hope the above paragraphs clarify the position of our paper in the context of Federated Learning.
> > >
> > >
> > > [1] Wen, Jie, et al. "A survey on federated learning: challenges and applications." International Journal of Machine Learning and Cybernetics 14.2 (2023): 513-535.
> > >
> > > Best,
> > >
> > > Authors

---

### Official Review · Reviewer_4ZYS · 2024-07-12

**Soundness:** 3
**Presentation:** 2
**Contribution:** 3
**Rating:** 5
**Confidence:** 3

**Summary:**

The paper presents a new federated learning approach named FEDNE for dimension reduction using contrastive neighbor embedding (NE). The key idea is the introduction of a surrogate loss function that each client learns and shares, which compensates for the lack of inter-client repulsion essential for global alignment in the embedding space. Additionally, the paper proposes a data-mixing strategy to augment local data, addressing issues of invisible and false neighbors in local kNN graphs. Comprehensive experiments demonstrate that FEDNE effectively preserves neighborhood data structures and enhances alignment in the global embedding space compared to several baseline methods.

**Strengths:**

1. The studied problem is important. There could be many downstream tasks after applying federated neighbor embedding.

2. Many metrics are included in the experiments to evaluate the quality of the resulting embeddings

**Weaknesses:**

1. The paper lacks investigation on the effect of choice of hyperparameter k.

2. The improvement of FEDNE is significant on some metrics (e.g., kNN) but is very limited in other metrics (e.g., conti.). The paper lacks a detailed exploration of why FEDNE produces different behavior for different metrics.

3. I suggest to highlight the best results in Table 2. Currently the results of FEDNE are highlighted although it may not achieve the best performance in some cases.

**Questions:**

1. How would the parameter k affect the performance of FEDNE? How to set k for different settings?

2. What are the major differences between the metrics? Why the improvement of FEDNE differ a lot across different metrics?

**Limitations:**

Yes.

---

> ### Author Rebuttal · Authors · 2024-08-07
>
> > How would the parameter k affect the performance of FEDNE? How to set k for different settings?
>
> Thank you for the valuable comment. First, we want to reiterate that the value k is used for building local kNN graphs to capture the neighboring data structures. In general, as k increases, we may lose the neighboring information. When k is too small, it may result in many isolated clusters in the embedding space.
>
> However, there is no clear consensus on how to select k in the dimensionality reduction (DR) community. As many recent DR papers use fixed k values [11], we followed them as well. We experimented with different k values under the setting of Dirichlet(0.1) on the MNIST dataset with 20 clients. **Please find the results in the attached PDF**.  We found that **within a certain range (i.e., 7 to 30), the performance of FedNE is relatively stable**. When k is too large (e.g., k=50), the performance drops but **our FedNE still outperforms the baseline methods**, FedAvg+NE. This trend aligns with the general understanding of DR methods. Once there is a best practice to select k for different settings, it can be applied to our problem.
>
> [11] “From t-SNE to UMAP with contrastive learning”. In ICLR, 2023.
>
>
> > What are the major differences between the metrics? Why the improvement of FEDNE differ a lot across different metrics?
>
> Thanks for your thoughtful comment. We had some discussions in Sections 5.1 and 5.2, but we should have made this more clear.
>
> Continuity, trustworthiness, and kNN classification accuracy are mainly used to evaluate the preservation of neighborhood data structures in the global reduced-dimensional space. Steadiness and cohesiveness aim to measure the preservation of inter-cluster structures since clusters can be distorted when projecting to the global low-dimensional space. Among the five metrics, trustworthiness, kNN classification accuracy, and steadiness have much more noticeable increases compared to continuity and cohesiveness.
>
> The improvement differs a lot among the metrics mainly because in a simple FedAvg framework without any special treatment for the data pairs across different clients, the model cannot penalize those data pairs to be apart, which results in introducing false neighbors in the embedding space (Line 215 and 292). These false neighbors can further cause data points from different classes to overlap in the embedding space. This explains why the baseline methods achieve much lower trustworthiness scores and kNN classification accuracy, as trustworthiness measures whether neighbors in the embedding space are also neighbors in the high-dimensional space.
>
> Due to similar reasons, without mitigating the incorrect neighbor connections and addressing the missing repulsion terms, the overlap in the embedding space will also mistakenly introduce false data clusters. Thereby, steadiness is also low for the baseline methods as it measures how well the model can avoid false clusters. However, both our surrogate models and intra-client data mixing strategy aim to prevent false neighbors in the embedding space. When true neighbors are preserved by FedNE, the trustworthiness and steadiness scores have increased and the points that are close in the embedding space are more likely to belong to the same class. Thus, the kNN classification accuracy is also notably increased (see Table 2 in the original manuscript).
>
> Continuity measures how well the neighborhood of a data point in the high-dimensional space is preserved in the embedding space. Achieving higher continuity is often easier even under the context of FL because the original attraction term in the client’s local objective function already pulls the neighbors closer in the embedding space. Cohesiveness is for measuring how well the projection model can avoid missing clusters. We still can observe relatively larger improvements in cohesiveness compared to continuity. Improved cohesiveness demonstrates that FedNE will not mistakenly break the true clusters in the embedding space.
>
>
> > I suggest to highlight the best results in Table 2. Currently the results of FEDNE are highlighted although it may not achieve the best performance in some cases.
>
> Thank you for the suggestion. We will change to highlight the best performance in the final version.

---

> > ### Comment · Reviewer_4ZYS · 2024-08-10
> >
> > Thanks for the authors' response. I'll keep my positive score.

---

> ### Author Response · Authors · 2024-08-12
> **Re: Official Comment by Reviewer 4ZYS**
>
> We appreciate your feedback and your positive opinion about our paper. If our rebuttal has addressed your concerns, we would also appreciate it if you would be willing to consider raising your original rating. Thank you for your consideration.

---

### Official Review · Reviewer_BJv7 · 2024-07-14

**Soundness:** 3
**Presentation:** 3
**Contribution:** 2
**Rating:** 5
**Confidence:** 3

**Summary:**

This paper addresses the challenge of distributed neural embedding (NE) with a focus on privacy protection. To achieve this, the authors extend the concept of federated learning (FL) to NE. However, NE tends to diverge because FL prevents clients from accessing each other's data, leading to inconsistent feature spaces across clients. To mitigate this issue, the authors employ surrogate loss models trained locally, which are then broadcast to all other clients to serve as an anchor. The experiments show promising performance compared to existing baselines.

**Strengths:**

1. The paper is well-motivated and well-written.
2. The problem is practical and useful for many real-life applications, though scalability may be the main constraint.
3. The idea is straightforward, and the experiments seem to verify its effectiveness.

**Weaknesses:**

1. **Communication complexity**: If I understand correctly, every client in the proposed method must broadcast the surrogate models to all other clients. Although the surrogate models consist of only one hidden layer, this design results in a communication complexity of $\mathcal{O}(N^2)$. As the number of clients in the system increases, the additional communication costs will rise dramatically. This might be manageable in some cross-silo settings, where only a few clients participate.

2. **Straggler effect**: Following point (1), the proposed method requires communication among clients. However, clients may drop out during training. It would be insightful if the authors could analyze how missing surrogate loss models would affect overall performance.

3. **Additional privacy concerns**: Sharing surrogate models introduces additional privacy risks, e.g., enabling reconstruction attacks or membership inference. While some recent work empirically shows that such private information is less leaked after distillation (e.g., [1] and [2]), the proposed method might be more vulnerable to privacy attacks without differential privacy.

[1] Dong, Tian, Bo Zhao, and Lingjuan Lyu. "Privacy for free: How does dataset condensation help privacy?." International Conference on Machine Learning. PMLR, 2022.
[2] Wang, Hui-Po, et al. "Fedlap-dp: Federated learning by sharing differentially private loss approximations," Proceedings on Privacy Enhancing Technologies, 2024.

**Questions:**

see weakness.

**Limitations:**

The authors have discussed some limitations. However, the authors are encouraged to discuss the use cases of the proposed method, such as cross-silo settings. Moreover, they are encouraged to discuss the additional privacy risks potentially introduced by their method.

---

> ### Author Rebuttal · Authors · 2024-08-07
>
> > Communication complexity ... this design results in a communication complexity of $O(N^2)$ … This might be manageable in some cross-silo settings, where only a few clients participate.
>
> Thanks for the thoughtful comment. Since each client will receive the surrogate models of all other clients from the server, we acknowledge that the size of the communication will be $O(N^2)$. We totally agree that our framework is more manageable in some cross-silo settings. However, as the size of the surrogate model is very small containing only one hidden layer (e.g., around 20K bytes), and nowadays data transfer has become much more efficient and affordable (e.g., 1.2 GB/s in modern wireless services), in our humble opinion, communication cost should not be a big problem.
>
> > Straggler effect … However, clients may drop out during training. It would be insightful if the authors could analyze how missing surrogate loss models would affect overall performance.
>
> Thank you for the nice practical question. We follow your suggestion to experiment with the idea that only a random 10% of the clients are involved in each communication round. Specifically, these clients will only receive the surrogate models from the other 10% of the clients. We summarize **our results in the attached PDF**, under the setting of Dirichlet(0.1) on the MNIST dataset with 100 clients. We can see that while the performance under 10% client participation is worse than under full client participation, **the results of FedNE are still notably better than the baseline methods, FedAvg+NE**, demonstrating the effectiveness and applicability of our FedNE framework.
>
> > Additional privacy concerns: Sharing surrogate models introduces additional privacy risks, … the proposed method might be more vulnerable to privacy attacks without differential privacy.
>
> Thank you so much for the comment. First, we want to apologize for the confusion caused by our title. The main focus of our paper is on a Federated setting, aiming to bring up the unique FL challenges and propose effective solutions for the pairwise training objective in the problem of Federated Neighbor Embedding. We appreciate your suggestions and will extend our discussion by adding a section in the camera-ready version to incorporate the points that you mentioned.
>
> Besides the privacy-preserving properties that a Federated setting has introduced, we do not specifically develop any privacy-preserving mechanisms. Therefore, we will surely refine our title and any writing to clarify this.
>
> With that being said, privacy-preserving is an important topic and we will explore incorporating differential privacy (or other techniques) in our future work. We will also investigate some characteristics of our algorithm to enhance privacy. For example, our surrogate models take the “low-dimensional” data as inputs. We will investigate whether this design improves privacy preservation since one cannot directly reconstruct the high-dimensional data. Moreover, since it is the server that collects surrogate models from all the clients and distributes them to each client, certain anonymization techniques may be applied so that each client does not know the owner of the surrogate function it receives.

---

> ### Comment · Reviewer_BJv7 · 2024-08-10
> **Response to the rebuttal**
>
> I want to thank the authors for their time and response. Since the response only partially addressed my concerns, I'd like to follow up on those points.
>
> > Scalability
>
> Despite the tiny surrogate model, the communication costs grow quadratically for the entire system and linearly for each client. It could be unfavorable when deployed in a million-scale system, but I can see the applications in common cross-silo settings.
>
> > Straggler effect
>
> Thanks for the additional experiments. However, I was curious about the stale surrogate models instead of partial participation. In other words, what would happen if some clients send old surrogate models from previous rounds due to the network delay?
>
> > Privacy concerns
>
> This remains my biggest concern. The authors do not provide any further analysis of it.
>
> While the concept of using shared anchors is not entirely new (as previously explored in [1] and preliminary theoretical analyses like [2]), its application in federated clustering appears to be new. The primary concern still lies in the potential privacy and security risks. I'm currently torn in my recommendation. On one hand, this work could provide a foundational analysis that serves as a baseline for federated clustering. On the other hand, the improved performance is built upon the inherent compromise of privacy. Such compromise also makes future comparisons unfair and challenging if the *threat model* is not stated clearly, i.e., *what is assumed to be safe*. Therefore, unless the potential privacy leakage is thoroughly discussed or the compatibility with differential privacy is demonstrated, I remain cautiously optimistic but will slightly lower my score.
>
> If the authors can propose a robust comparison protocol for future work that accounts for various privacy considerations, I would be willing to reconsider and potentially adjust my score.
>
> [1] Wang, Hui-Po, et al. "Fedlap-dp: Federated learning by sharing differentially private loss approximations," Proceedings on Privacy Enhancing Technologies, 2024.\
> [2] Li, Bo, et al. "Synthetic data shuffling accelerates the convergence of federated learning under data heterogeneity." Transactions on Machine Learning Research.

---

> > ### Author Response · Authors · 2024-08-11
> >
> > Dear reviewer,
> >
> > Thank you for reading our rebuttal and we apologize that our rebuttal has not fully addressed your concerns. We respond to your remaining concerns as follows.
> >
> > We totally agree that our framework is more manageable in some cross-silo settings. Speaking of the communication cost on extremely large systems e.g., a million-scale system, this is still an unresolved problem in the existing literature. We have made one step further to relax the assumption of publicly available data and the next step in FL for NE could be reducing the communication cost.
> >
> >
> > Regarding the potential stale/old surrogate models due to network delay, we had a related experiment in Appendix B.2 where we studied whether reducing the frequency of surrogate model updates would impact the final performance (i.e., under the Shards setting with 20 clients on MNIST), even though in this experiment we assume that all clients may send out their outdated/stale surrogate models instead of only some of the clients. We can see that **the performance remains stable** and **the improvement is still notable compared to the baseline method, FedAvg+NE**, if the surrogate models are only outdated within a threshold of iterations i.e., 10 rounds of communication (Fig.4 in Appendix). While we know this experiment does not exactly match what you asked for, we hope it can still provide a better understanding of our method.
> >
> >
> > Lastly, we do agree that *privacy is a very important problem*. After serious consideration, there are several potential future directions on how to address and evaluate the privacy considerations in our framework. As demonstrated in Section 4 and the workflow in Figure 1, FedNE is built upon the traditional FedAvg framework. The potential privacy concerns mainly come from the FedAvg framework and the surrogate models that we proposed.
> >
> > To address the **privacy risks associated with FedAvg**, one potential solution is to implement a FedAvg version of DP-SGD [1]. In this approach, the Gaussian mechanism (GM) is applied to the global model updates, as described in line 10 of Algorithm 1 in [1], to achieve client-level differential privacy (DP). That is, each client’s entire dataset is protected against differential attacks from other clients.
> >
> > To mitigate the **privacy concerns associated with** our proposed **surrogate models**, it is important to both anonymize client identities and address the potential risks inherent in sharing these models. To make the surrogate models differentially private, DP techniques can be integrated at different stages. One way is to incorporate DP before sharing the surrogate models. Even though the surrogate models are simple and only contain highly compressed information about the client data, the privacy risks can be further mitigated by applying the Gaussian mechanisms (GM) to the parameters of the surrogate model before it is sent to the server. Additionally, DP techniques can be incorporated during the training phase of the surrogate models through data synthesis. Specifically, to prepare the training targets, instead of using real data, each client can generate synthetic samples (e.g., via data distillation) to compute the repulsion loss values $l_q^{rep}$​ (as discussed in Lines 198-201), which are then used to train the surrogate models.
> >
> > To ensure privacy guarantees, when GM is employed, a desired privacy budget can be set using Epsilon ($\epsilon$) and Delta ($\delta$), while balancing between privacy and utility. The moments accountant method can be utilized to track cumulative privacy loss over multiple iterations of FedNE. Furthermore, the privacy auditing algorithm [2] provides a valuable tool for estimating empirical privacy lower bounds within our framework. **The empirical Epsilon can be an indicator** (i.e., a lower Epsilon, a stronger privacy guarantee) **for comparing different methods**, e.g., FedAvg, FedNE, even FedNE integrated with DP, and other future works.
> >
> > [1] Geyer, Robin C., Tassilo Klein, and Moin Nabi. "Differentially private federated learning: A client level perspective." arXiv preprint arXiv:1712.07557 (2017).
> >
> > [2] Steinke, Thomas, Milad Nasr, and Matthew Jagielski. "Privacy auditing with one (1) training run." Advances in Neural Information Processing Systems 36 (2024).
> >
> > Please kindly let us know if you have any further questions or concerns. We are more than happy to have a further discussion regarding it.
> >
> > Best,
> >
> > Authors

---

> ### Author Response · Authors · 2024-08-12
> **Kindly request your reconsideration**
>
> Dear Reviewer BJv7,
>
> We appreciate your timely feedback on our rebuttal. Given the limited author-reviewer discussion period, we have tried our best to further address it. Please see our response titled "Official Comment by Authors." If our latest response has addressed your concerns, we would appreciate it if you would be willing to consider raising your rating. Thank you for your consideration.
>
> Best,
>
> Authors

---

> > ### Comment · Reviewer_BJv7 · 2024-08-13
> >
> > Dear Authors,
> >
> > Thank you for your prompt response. I’ve carefully reviewed your feedback, and I’m glad to see that the authors agree on the privacy guarantee aspect. While I appreciate the explanation of the common practices in implementing differential privacy (DP), my primary concern remains that the paper might set an unfair baseline for future work if proper DP experiments are not included. Therefore, I will be maintaining my score.

---

> > > ### Author Response · Authors · 2024-08-14
> > >
> > > Dear Reviewer BJv7,
> > >
> > > Thank you for your timely response and further clarification of your concerns. We sincerely apologize if we have misunderstood your original review and your response to the rebuttal (on 08/10/2024). If we read them correctly, you encouraged us to **discuss** the additional privacy risks potentially introduced by their method, and **propose** a robust comparison protocol for future work that accounts for various privacy considerations. We have tried our best to follow your suggestions in our rebuttal and additional responses. However, it seems that your latest comment demands us to further provide **experiments with DP** to address your concern. While we will be happy to do so, we, unfortunately, cannot complete it on the last day of the discussion phase. Still, we sincerely thank you for all your constructive feedback and suggestions.
> > >
> > >
> > > Best,
> > >
> > > Authors

---

### Author Rebuttal · Authors · 2024-08-07

We thank the reviewers for all the valuable comments and constructive suggestions. We are glad that the reviewers found that our paper is “well-motivated” and “well-presented” (Reviewer BJv7, 4ZYS, mP1Q), and our approach is “novel” (Reviewer mP1Q, y2ZN).

In the following, we want to first reiterate our contributions. We then respond to each individual reviewer’s comments separately. To address some of the questions, we have included additional experimental results in the attached PDF.  We will also incorporate all the feedback in the camera-ready version.

### **Overall contributions and setting**

We want to emphasize that we study a less-explored problem in Federated Learning (FL), FL for Neighbor Embedding (NE), where NE is a family of non-linear Dimensionality Reduction (DR) techniques. As discussed in Section 3.3, in comparison to classification tasks, FL for NE has a unique challenge associated with the pairwise objective function in the NE problem.

One of our key contributions is to point out these challenges systematically and provide effective solutions. We have used the benchmark datasets commonly studied in existing DR literature, and our experiments are larger in scale compared to prior work. We appreciate some reviewers’ feedback regarding complexity and scalability, and we have tried our best to respond to them.

Overall, there are many properties/challenges that one would expect a sophisticated FL algorithm to achieve and overcome. The area of FL has been significantly advanced in the past five years, thanks to hundreds, if not thousands, of papers addressing various aspects of FL. In this paper, we follow this trend by identifying a new challenge specific to FL for NE and dedicating ourselves to mitigating it. Thus, while our algorithm may not be as computationally efficient or scalable as some existing FL methods, we respectfully believe that our contributions are significant and valuable for future study in the field of FL for DR.


### **Regarding scalability, complexity, and computational efficiency**

Our paper focuses on a less-explored combination of FL and DR. Please allow us to address the questions regarding scalability, complexity, and computational efficiency in three aspects.

First, to our knowledge, DR has focused on relatively smaller-scale datasets, compared to classification. This is because computational complexity is never a trivial problem even for many outstanding DR techniques, particularly for non-linear methods such as Isomap and t-SNE which have non-convex cost functions [44]. Indeed, our experiments have included the most widely-used benchmarks used in the DR literature.

Second, in terms of FL, we work on a less-explored problem, compared to other learning problems such as classification. Compared to the latter, in which the instance-based loss can closely approximate the centralized loss with frequent communication (Line150-154), the Federated Neighbor Embedding problem has a unique challenge associated with its pairwise objective function. Thus, we focus our study mostly on addressing this challenge using standard DR datasets. With that being said, compared to prior work on decentralized data projection, we already considered more clients and larger datasets. Specifically, d-SNE [37] and F-dSNE [38] conducted experiments with *only 3 to 10 local sites* on the *subsampled* MNIST and biomedical datasets which contain *only hundreds to thousands* of data samples. However, our FedNE is evaluated on the *full benchmark datasets* with the settings of *100 local sites*.

Third, while we have not studied larger datasets and more clients like other FL works in classification, we expect that our approach is applicable in real-world settings, for example, cross-silo settings with a manageable amount of clients. As pointed out in Sect. 4.3 (Line 235-237), our approach only requires 35% additional GPU time compared to FedAvg, and we expect such overhead to remain similar when going to larger datasets with a similar number of clients. When the number of clients increases, we may optionally drop a portion of surrogate models in local training. As shown in our response to Reviewer BJv7, such a setting does not lead to a significant performance drop but can maintain scalability.


[37] "See without looking: joint visualization of sensitive multi-site datasets." IJCAI. 2017.

[38] "Federated, Fast, and Private Visualization of Decentralized Data." Workshop of Federated Learning and Analytics in Practice (2023).

[44] "Visualizing data using t-SNE." JMLR 9.11 (2008).

---

### Comment · Area_Chair_T8i3 · 2024-08-12
**Paper discussion**

Hi all,

Thanks again for your reviews!
If you haven't done so already, please respond to the rebuttals before the end of the author-reviewer discussion period. If you don't have any further questions for the authors then a simple "thank you for the rebuttal" would suffice.

All the best,
Area Chair

---

### Decision · Program_Chairs · 2024-09-25

**Decision:**

Accept (poster)

**Comment:**

The paper presents a new federated learning approach for dimension reduction using contrastive neighbor embedding (NE). The key idea is the introduction of a surrogate loss function that each client learns and shares, which compensates for the lack of inter-client repulsion essential for global alignment in the embedding space. Additionally, the paper proposes a data-mixing strategy to augment local data, addressing issues of invisible and false neighbors in local kNN graphs. Comprehensive experiments demonstrate that FEDNE effectively preserves neighborhood data structures and enhances alignment in the global embedding space compared to several baseline methods.

1.	FEDNE introduces a novel integration of FEDAVG with contrastive NE techniques, addressing the unique challenges of pairwise data relationships in federated learning environments without requiring data sharing.

2.	The intra-client data mixing strategy effectively enhances local data diversity, mitigating the limitations of biased local kNN graphs and ensuring better neighborhood representation.

3.	The paper provides a thorough evaluation of FEDNE using various datasets and metrics, showcasing its superior performance compared to baseline methods in preserving neighborhood structures and clustering.

Weaknesses:
The title should not include the “privacy-preserving” as it doesn’t show additional privacy guarantee besides federated learning itself.

1.	Communication complexity: The method appears to be computationally inefficient. There are no experiments conducted on large datasets, such as those in real-world medical or other privacy-critical domains, where computational complexity could be a significant issue.

2.	Additional privacy concerns: Sharing surrogate models introduces additional privacy risks, e.g., enabling reconstruction attacks or membership inference. While some recent work empirically shows that such private information is less leaked after the proposed method might be more vulnerable to privacy attacks without differential privacy. It lacks experiments and guarantees demonstrating the privacy preservation of the FedNE approach. The improved performance is built upon the inherent compromise of privacy. Such compromise also makes future comparisons unfair and challenging if the threat model is not stated clearly, i.e., what is assumed to be safe.